# Low doses of the organic insecticide spinosad trigger lysosomal defects, elevated ROS, lipid dysregulation, and neurodegeneration in flies

Felipe Martelli[1†], Natalia H Hernandes[1], Zhongyuan Zuo[2], Julia Wang[2‡], Ching-On Wong[3§], Nicholas E Karagas[3], Ute Roessner[1], Thusita Rupasinghe[1], Charles Robin[1], Kartik Venkatachalam[3], Trent Perry[1], Philip Batterham[1], Hugo J Bellen[2,4,5]*

[1]School of BioSciences, The University of Melbourne, Melbourne, Australia; [2]Department of Molecular and Human Genetics, Baylor College of Medicine, Houston, United States; [3]Department of Integrative Biology and Pharmacology, McGovern Medical School at the University of Texas Health Sciences Center, Houston, United States; [4]Neurological Research Institute, Texas Children Hospital, Houston, United States; [5]Howard Hughes Medical Institute, Baylor College of Medicine, Houston, United States

*For correspondence: hbellen@bcm.edu

Present address: †School of Biological Sciences, Monash University, Melbourne, Australia; ‡Medical Scientist Training Program, Baylor College of Medicine, Houston, United States; §Department of Biological Sciences, Rutgers University, Newark, United States

**Abstract** Large-scale insecticide application is a primary weapon in the control of insect pests in agriculture. However, a growing body of evidence indicates that it is contributing to the global decline in population sizes of many beneficial insect species. Spinosad emerged as an organic alternative to synthetic insecticides and is considered less harmful to beneficial insects, yet its mode of action remains unclear. Using *Drosophila*, we show that low doses of spinosad antagonize its neuronal target, the nicotinic acetylcholine receptor subunit alpha 6 (nAChRα6), reducing the cholinergic response. We show that the nAChRα6 receptors are transported to lysosomes that become enlarged and increase in number upon low doses of spinosad treatment. Lysosomal dysfunction is associated with mitochondrial stress and elevated levels of reactive oxygen species (ROS) in the central nervous system where nAChRα6 is broadly expressed. ROS disturb lipid storage in metabolic tissues in an nAChRα6-dependent manner. Spinosad toxicity is ameliorated with the antioxidant N-acetylcysteine amide. Chronic exposure of adult virgin females to low doses of spinosad leads to mitochondrial defects, severe neurodegeneration, and blindness. These deleterious effects of low-dose exposures warrant rigorous investigation of its impacts on beneficial insects.

## Editor's evaluation

Insecticides have been implicated in the decline of beneficial insect species. The organic insecticide Spinosad has emerged as a alternative to synthetic insecticides and is thought to be less harmful to beneficial insects than synthetic insecticides. This article used the insect model Drosophila to analyze the impact of Spinosad. It reveals that low doses of Spinosad antagonize its neuronal target, the nicotinic acetylcholine receptor subunit alpha 6, affecting the physiology of Drosophila. This study reveals that although being organic and assumed to be less harmful, spinosad can have profound impact on non-target insects.

## Introduction

Insecticide applications maximize crop yield, but negatively impact populations of insects that provide beneficial services in agriculture and horticulture (*Sánchez-Bayo and Wyckhuys, 2019*). The global decline in population sizes of these beneficial insects creates challenges for ecosystems and farming. Although estimates differ depending on the regions and the methodologies used (*Wagner et al., 2021*), one recent study suggests an approximately 9% decline in terrestrial insect abundance per decade since 1925 (*van Klink et al., 2020*). While the precise extent to which insecticides are involved remains undetermined, they have consistently been associated as a key factor, along with climate change, habitat loss, and increased levels of pathogens and parasites (*Cardoso et al., 2020*; *Sánchez-Bayo and Wyckhuys, 2019*; *Wagner et al., 2021*). Much attention has been given to neonicotinoid insecticides, both in the scientific literature and in public discourse, because of the evidence that these chemicals contribute to the bee colony collapse phenomenon (*Lu et al., 2014*; *Lundin et al., 2015*).

In assessing the risk posed by insecticides, it is important to study the molecular and cellular events that unfold following the interaction between the insecticide and its target. Many insecticides target ion channels in the nervous system. At the high doses used to kill pests, these insecticides produce massive perturbations to the flux of ions in neurons, resulting in lethality (*Breer and Sattelle, 1987*; *Perry and Batterham, 2018*; *Scott and Buchon, 2019*). But non-pest insects are likely to be exposed to much lower doses, and the downstream physiological processes that are triggered are poorly understood. In a recent study, low doses of the neonicotinoid imidacloprid were shown to stimulate an enduring flux of calcium into neurons via the targeted ligand-gated ion channels (nicotinic acetylcholine receptors [nAChRs]) (*Martelli et al., 2020*). This causes an elevated level of reactive oxygen species (ROS) and oxidative damage that radiates from the brain to other tissues. Mitochondrial stress leads to a significant drop in energy levels, neurodegeneration, and blindness (*Martelli et al., 2020*). Evidence of compromised immune function was also presented, supporting other studies (*Chmiel et al., 2019*). Many other synthetic insecticides are known to elevate the levels of ROS (*Karami-Mohajeri and Abdollahi, 2011*; *Lukaszewicz-Hussain, 2010*; *Wang et al., 2016*) and may precipitate similar downstream impacts. Given current concerns about synthetic insecticides, a detailed analysis of the molecular and cellular impacts of organic alternatives is warranted. Here, we report such an analysis for an insecticide of the spinosyn class, spinosad.

Spinosad is an 85%:15% mixture of spinosyns A and D, natural fermentation products of the soil bacterium *Saccharopolyspora spinosa*. It occupies a small (3%) but growing share of the global insecticide market (*Sparks et al., 2017*). It is registered for use in more than 80 countries and applied to over 200 crops to control numerous pest insects (*Biondi et al., 2012*). Recommended dose rates vary greatly depending on the pest and crop, ranging from 96 parts per million (ppm) for Brassica crops to 480 ppm in apple fields (*Biondi et al., 2012*). Like other insecticides, the level of spinosad residues found in the field varies greatly depending on the formulation, the application mode and dose used, environmental conditions, and proximity to the site of application. If protected from light, spinosad shows a half-life of up to 200 days (*Cleveland et al., 2002*).

Spinosad is a hydrophobic compound belonging to a lipid class known as polyketide macrolactones. Studies using mutants, field-derived-resistant strains, and heterologous expression have shown that spinosad targets the highly conserved nAChRα6 subunit of nAChRs in *Drosophila melanogaster* (hereafter Dα6) and a range of other insect species (*Perry et al., 2015*; *Perry et al., 2007*). Spinosad is an allosteric modulator, binding to a site in the C terminal region of the protein (*Puinean et al., 2013*; *Somers et al., 2015*). *Salgado and Saar, 2004* found that spinosad allosterically activates non-desensitized nAChRs, but that low doses were also capable of antagonizing the desensitized nAChRs. It is currently accepted that spinosad causes an increased sensitivity to ACh in certain nAChRs and an enhanced response at some GABAergic synapses, causing involuntary muscle contractions, paralysis, and death (*Biondi et al., 2012*; *Salgado, 1998*). However, a recent study (*Nguyen et al., 2021*) showed that both acute and chronic exposures to spinosad cause Dα6 protein levels in the larval brain to decrease. A rapid loss of Dα6 protein during acute exposure was blocked by inhibiting the proteasome system (*Nguyen et al., 2021*). As *Dα6* loss-of-function mutants are both highly resistant to spinosad and viable (*Perry et al., 2021*; *Perry et al., 2007*), it was suggested that the toxicity of spinosad may be due to the overloading of protein degradation pathways and/or the internalization of spinosad where it may cause cellular damage. Higher doses of spinosad than the ones used here have been shown to cause cellular damage via mitochondrial dysfunction, oxidative stress, and

programmed cell death in cultured insect cells (*Spodoptera frugiperda* Sf9) (*Xu et al., 2017*; *Yang et al., 2017*).

Here, we show that spinosad by itself does not increase $Ca^{2+}$ flux in *Drosophila* neurons. Indeed, the response elicited by a cholinergic agonist is stunted upon spinosad treatment. Following exposure to spinosad, Dα6 cholinergic receptors are endocytosed and trafficked to the lysosomes, leading to lysosomal dysfunction. This dysfunction is associated with high levels of oxidative stress. Antioxidant treatment prevents the accumulation of ROS, but not lysosomal expansion. ROS is a key factor in the mode of action of spinosad at low doses, triggering a cascade of damage that results in mitochondrial stress and reduced energy levels. Low chronic exposures lead to extensive neurodegeneration in the central brain and blindness. Flies carrying a *Dα6* loss-of-function mutation show a mild increase in ROS, but no evidence of lysosomal dysfunction. This indicates that the lysosomal defect observed in wild-type flies is not due to the absence of Dα6 from neuronal membranes but rather trafficking of Dα6 to lysosomes under conditions of spinosad exposure. Given the high degree of conservation of the spinosad target between insect species (*Perry et al., 2015*), our data indicate that this insecticide has the potential to cause harm in non-pest insects at low doses.

## Results

### Low doses of spinosad affect survival and prevent $Ca^{2+}$ flux into neurons expressing Dα6

As a starting point to study the systemic effects of low-dose spinosad exposure, a dose that would reduce the movement of third-instar larvae by 50% during a 2 hr exposure was determined. This was achieved with a dose of 2.5 ppm (*Figure 1A*). Under this exposure condition, only 4% of wild-type larvae survived to adulthood (*Figure 1B*), whereas 88% *nAChRα6 knockout* (*Dα6 KO*) mutants survived (*Figure 1C*). The effect of this dose was measured on cultured primary neurons of third-instar larva brain, where the *Dα6* gene promoter was used to drive the GCaMP5G:tdTomato cytosolic [$Ca^{2+}$] sensor. As no alterations in basal $Ca^{2+}$ levels were detected in neurons expressing *Dα6* response to 2.5 ppm (*Figure 1D and E*), a dose of 25 ppm was tested, again with no measurable impact (*Figure 1D and E*). After 5 min of spinosad exposure, neurons were stimulated by carbachol, a cholinergic agonist that activates nAChR. Spinosad-exposed neurons exhibited a significant decrease in cholinergic response when compared to unexposed neurons (*Figure 1D and E*). Total $Ca^{2+}$ content mobilized from ER remained unaltered as measured by thapsigargin-induced $Ca^{2+}$ release (*Figure 1D and E*). While it was not determined whether the $Ca^{2+}$ transients reflect reduced influx from internal or external sources (*Campusano et al., 2007*), spinosad exposure led to a diminished $Ca^{2+}$ transient and reduced cholinergic response. Hence, in contrast to imidacloprid, which leads to an enduring $Ca^{2+}$ influx in neurons (*Martelli et al., 2020*), spinosad reduces the $Ca^{2+}$ response mediated by Dα6.

### Spinosad exposure causes lysosomal dysfunction in a Dα6-dependent manner

Spinosad exposures cause a gradual reduction in the Dα6 signal in brains (*Figure 2A and B*; *Nguyen et al., 2021*). To test whether spinosad affects lysosomes, we stained larval brains with LysoTracker. No phenotype was observed after 1 hr exposure, but after a 2 hr exposure at 2.5 ppm, spinosad caused an eightfold increase in the area occupied by lysosomes (*Figure 2C and E*). 6 hr after the 2 hr initial exposure ended, the area occupied by lysosomes in brains was 24-fold greater than in unexposed larvae (*Figure 2C and E*). In contrast, no increased number of enlarged lysosomes were observed in *Dα6 KO* mutants in the presence or absence of spinosad exposure (*Figure 2D and F*). These data indicate that the lysosomal expansion is dependent on both the presence of the Dα6 receptors and spinosad. Significantly, the Dα6 receptors were found to colocalize with the enlarged lysosomes (*Figure 2G*), indicating that enlarged lysosomes are trapping Dα6 receptors in response to spinosad exposure.

### Spinosad exposure affects mitochondria turnover and reduces energy metabolism that is counteracted by antioxidant treatment

Defects in lysosomal function have been shown to impact other organelles, especially mitochondria (*Deus et al., 2020*). To investigate whether mitochondrial function was also affected by spinosad

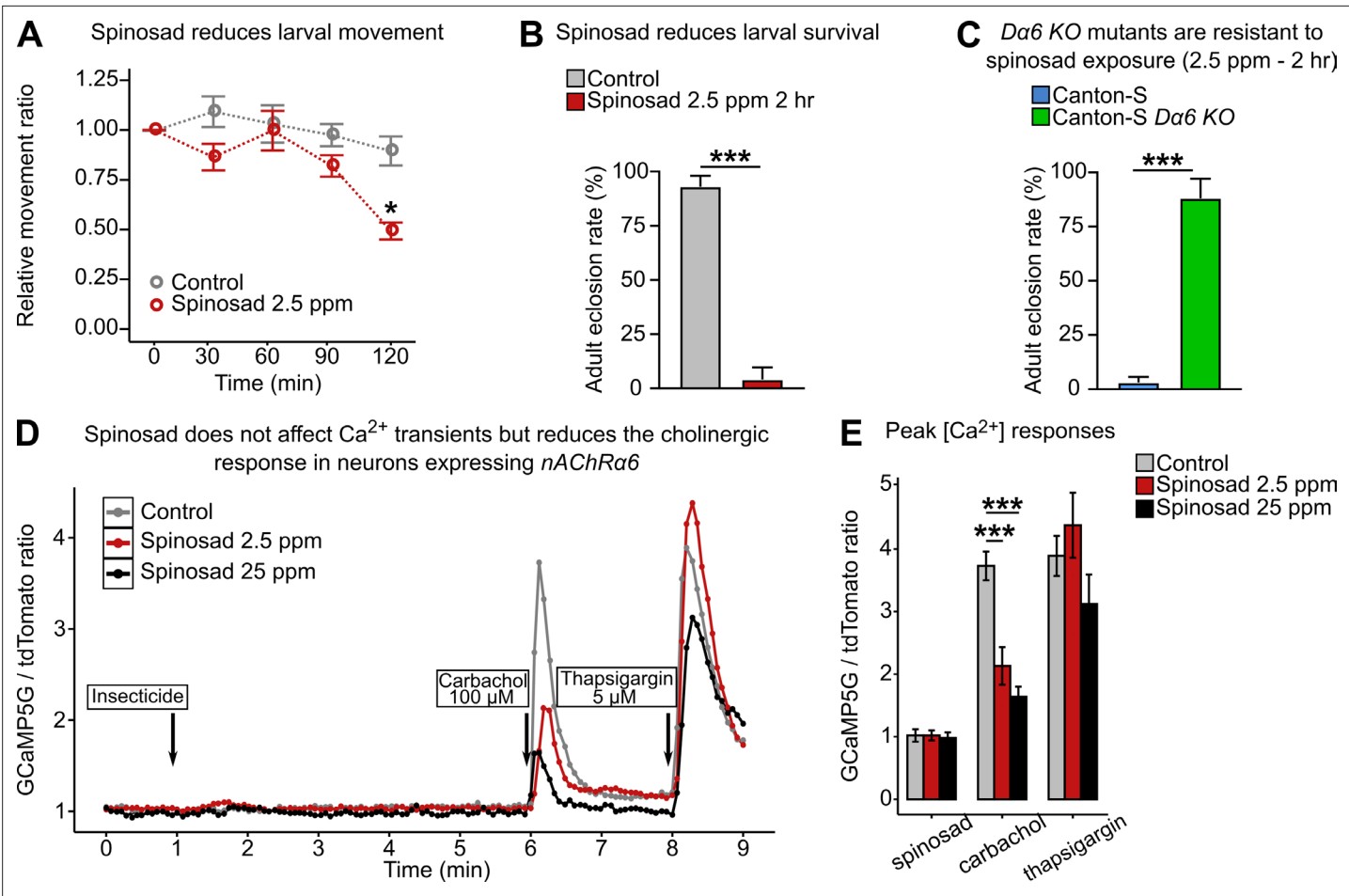

**Figure 1.** Low doses of spinosad are lethal and fail to increase Ca$^{2+}$ levels in neurons. (**A**) Dose–response to spinosad in Line14 wild-type larvae by an assay of larval movement over time, expressed in terms of relative movement ratio (n = 100 larvae/treatment). (**B**) Adult eclosion rate after Line14 larvae were subjected to a 2 hr exposure to 2.5 parts per million (ppm) spinosad, rinsed and placed back onto insecticide-free medium (n = 100 larvae/treatment). (**C**) Adult eclosion rate of Canton-S and Canton-S Dα6 KO larvae subjected to a 2 hr exposure at 2.5 ppm spinosad, rinsed and placed back onto insecticide-free medium (n = 100 larvae/treatment). (**D**) Cytosolic [Ca$^{2+}$] measured by GCaMP in neurons expressing Dα6. Measurement is expressed as a ratio of the signals of GCaMP5G signal and tdTomato. Spinosad (2.5 ppm or 25 ppm) was added to the bath solution at 1 min after recording started. At 6 min and 8 min, the spinosad-exposed and unexposed groups were stimulated with 100 μM carbachol and 5 μM thapsigargin, respectively. Each point represents the average of at least 50 cells. (**E**) Peak [Ca$^{2+}$] responses to spinosad and carbachol. Error bars in (**A**) and (**E**) represent mean ± SEM and in (**B**) and (**C**) mean ± SD. (**A, E**) One-way ANOVA followed by Tukey's honestly significant difference test; (**B, C**) Student's unpaired $t$-test. *$p < 0.05$, ***$p < 0.001$.

exposure, we assessed mitochondrial turnover using the MitoTimer reporter line (*Laker et al., 2014*). A 2 hr spinosad exposure induced an increase of 31% and 36% for the green (healthy mitochondria) and red (stressed mitochondria) signals in the optic lobes of the larval brain, respectively (*Figure 3A and B*). For the digestive tract, a 19% and 32% increase was observed in the proventriculus for green and red signals, respectively (*Figure 3A and C*). The mito-roGFP2-Orp1 construct, an in vivo mitochondrial H$_2$O$_2$ reporter (*Albrecht et al., 2011*), was used to identify the origin of ROS induced by spinosad exposure at 2.5 ppm for 2 hr. A subtle, but significant, increase in the 405 (oxidized mitochondria signal)/488 (reduced mitochondria signal) ratio was observed in the brain (20% on average) and anterior midgut (10% on average) (*Figure 3—figure supplement 1*), pointing to a rise in H$_2$O$_2$ generation upon a few hours of exposure. Similarly to the MitoTimer reporter, an increase in the oxidized mitochondrial signal was accompanied by the increase in the reduced mitochondrial signal, accounting for the subtle increase in 405/488 ratio. To further examine whether the results obtained with the mitochondrial reporters were connected to increased ROS production in mitochondria, we measured the enzyme activity of mitochondrial aconitase (m-aconitase), a highly ROS-sensitive

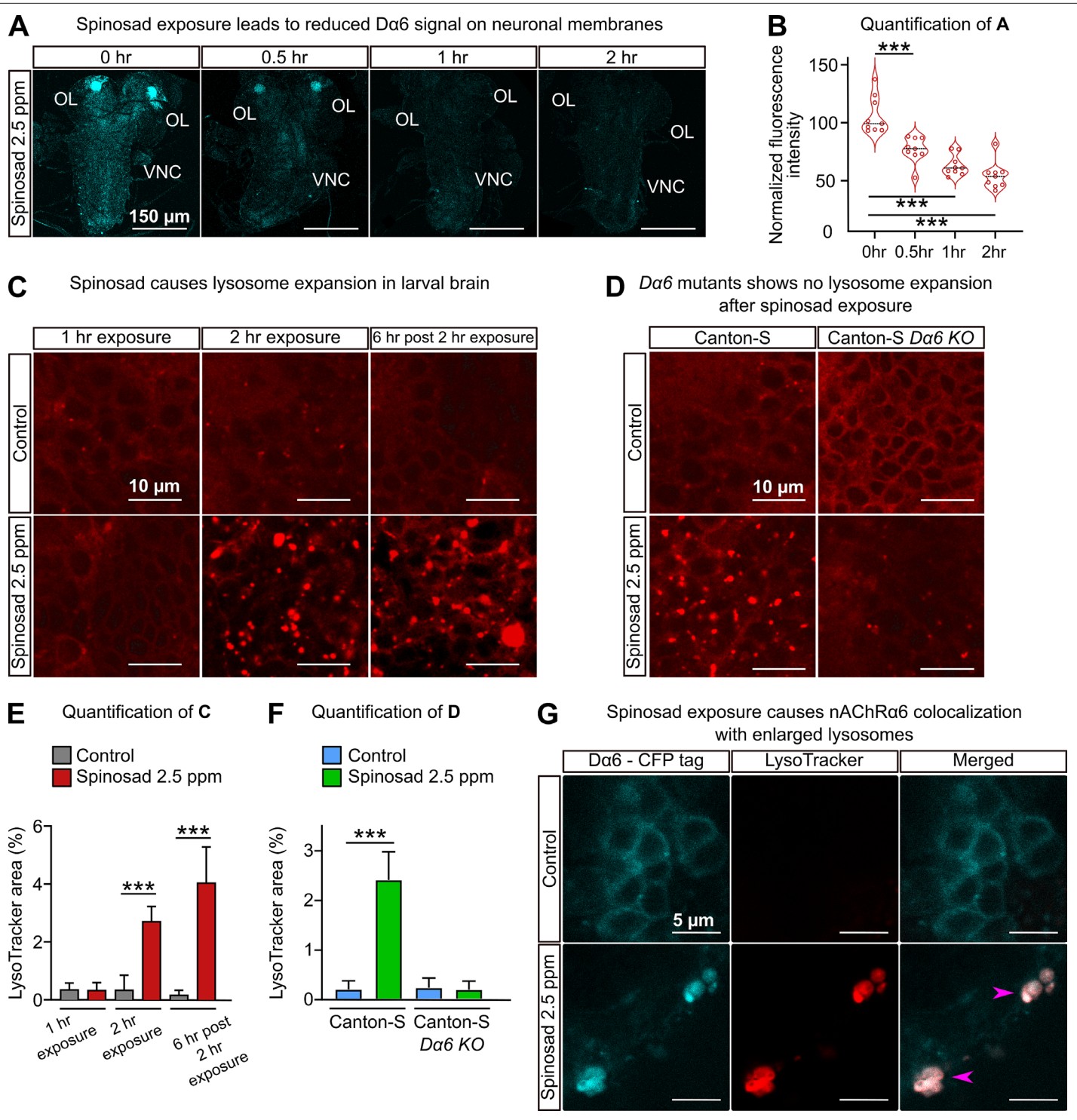

**Figure 2.** Spinosad exposure causes lysosomal expansion, and Dα6 colocalizes with enlarged lysosomes. (**A**) Dα6 signal in larval brains exposed to 2.5 parts per million (ppm) spinosad for 30 min, 1 hr, or 2 hr. Larvae obtained by crossing UAS-Dα6 (CFP tagged) in the Line14 Dα6nx loss-of-function mutant background to a native Dα6 driver (the Gal4-L driver) in the same background. OL, optic lobe; VNC, ventral nerve cord. (**B**) Quantification of (**A**) (n = 3 larvae/condition; three brain sections/larva). (**C**) LysoTracker staining shows lysosome expansion in the brain of Line14 larvae exposed to 2.5 ppm spinosad for 1 hr, 2 hr, or 6 hr post the 2 hr exposure. (**D**) LysoTracker staining shows lysosomes in the brain of Canton-S and Canton-S Dα6 KO larvae exposed to 2.5 ppm spinosad for 2 hr. (**E**) Quantification of (**C**), LysoTracker area (%) (n = 7 larvae/treatment; three optic lobe sections/larva). (**F**) Quantification of (**D**), LysoTracker area (%) (n = 7 larvae/treatment; three optic lobe sections/larva). (**G**) Larvae expressing Dα6 tagged with CFP exposed to 2.5 ppm spinosad for 2 hr show colocalization of the Dα6 and lysosomal signals. Pink arrowheads indicate Dα6 CFP signal colocalizing with

*Figure 2 continued on next page*

*Figure 2 continued*

lysosomes identified with LysoTracker staining. Microscopy images were obtained with a Leica SP5 Laser Scanning Confocal Microscope. Error bars in (**E**) and (**F**) represent mean ± SD. (**B**, **E**, **F**) One-way ANOVA followed by Tukey's honestly significant difference test. ***p<0.001.

enzyme (*Yan et al., 1997*). We observed a mean 34% reduction in m-aconitase activity (*Figure 3D*), indicating an increased presence of ROS in mitochondria after the 2 hr exposure. Immediately after the 2 hr exposure, a mean 36% increase in systemic ATP levels was observed, followed by a 16.5% reduction 12 hr after the 2 hr exposure (*Figure 3E*). The initial increase in energy levels is consistent with the increase in the signals of healthy and stressed mitochondria identified by the MitoTimer and mito-roGFP reporters at this time point. However, the reduction in ATP levels 12 hr after the exposure shows that the mitochondrial energy output is eventually impaired.

To quantify the extent to which oxidative stress generated by 2.5 ppm spinosad exposure for 2 hr could affect larval motility and survival, larvae were treated with the antioxidant N-acetylcysteine amide (NACA) (*Schimel et al., 2011*) for 5 hr prior to insecticide exposure. NACA treatment improved larval motility by ~50% at the 2 hr exposure time point when compared to larvae not treated with the antioxidant (*Figure 3F*). Adult eclosion rates increased from an average 4% to 15% when larvae exposed to spinosad were treated with NACA (*Figure 3F*). These results show a causal role for oxidative stress in the mode of action of spinosad at low doses. Nonetheless, the results also suggest that oxidative stress is not the only responsible mediator for the detrimental effects of spinosad exposure. Cross-talk between mitochondrial stress and lysosome dysfunction may be the major culprit for the highly toxic effects of low levels of spinosad exposure. This relationship is further investigated below.

## Antioxidant treatment prevents ROS accumulation but not lysosomal expansion

Given the evidence for increased ROS production, we next examined the levels of the anion $O_2^-$ (superoxide), a primary ROS produced by mitochondria (*Valko et al., 2007*), as well as other ROS sources (*Zielonka and Kalyanaraman, 2010*), using dihydroethidium (DHE) staining. After a 1 hr exposure to 2.5 ppm spinosad, there was a mean 89% increase of ROS levels in the brain. After 2 hr, the levels were lower than that at the 1 hr time point, but still 44% higher than that in the unexposed controls (*Figure 4A and B*). A different pattern was observed in the anterior midgut. A significant increase of ROS levels compared with the controls (28%) was observed only at the 2 hr time point (*Figure 4A and C*). Unexposed *Dα6 KO* mutants showed a mild (17%) increase in ROS levels in the brain when compared to unexposed wild-type larvae (*Figure 4—figure supplement 1*). Exposure to spinosad caused no alteration of ROS levels in *Dα6 KO* mutants (*Figure 4—figure supplement 1*). Hence, the absence of Dα6 subunit by itself is able to modestly increase the oxidative stress (*Weber et al., 2012*), but higher levels of ROS are only observed in the presence of Dα6 and spinosad. To assess the mitochondrial origin of the ROS measured with DHE, flies expressing the superoxide dismutase gene *Sod2* in the nervous system with the *elav-GAL4* driver were exposed to 2.5 ppm spinosad for 2 hr. Sod2 is the main ROS scavenger in *Drosophila* and is localized to mitochondria (*Missirlis et al., 2003*). Sod1 is present in the cytosol (*Missirlis et al., 2003*), and expression of this gene was used as a control. While exposure to spinosad caused an average 63% increase in ROS levels in control larvae overexpressing *Sod1*, an average increase of only 28% was found in larvae overexpressing *Sod2* (*Figure 4D and E*).

To further dissect the relationship between lysosome dysfunction and mitochondrial stress, we exposed larvae treated with NACA to spinosad and once again quantified the levels of ROS and the area covered by lysosomes in brains. Whereas NACA treatment was able to prevent ROS accumulation in exposed animals (*Figure 4F and G*), it did not prevent lysosome expansion (*Figure 4H and I*). The presence of enlarged lysosomes in the absence of ROS suggests that the onset of the lysosomal phenotype is not caused by the rise in oxidative stress levels. NACA, however, reduced the severity of the lysosomal phenotype (mean 1.63% of lysotracker area [*Figure 4I*] versus mean 2.39% of lysotracker area [*Figure 2E*]). This suggests that, once initiated, the increase in ROS levels may worsen the phenotype associated with lysosomal dysfunction.

## Brain signals trigger the impact of spinosad on metabolic tissues

Oxidative stress has the ability to affect the lipid environment of metabolic tissues, causing bulk redistribution of lipids into lipid droplets (LDs) (*Bailey et al., 2015*). The RNAi knockdown of mitochondrial

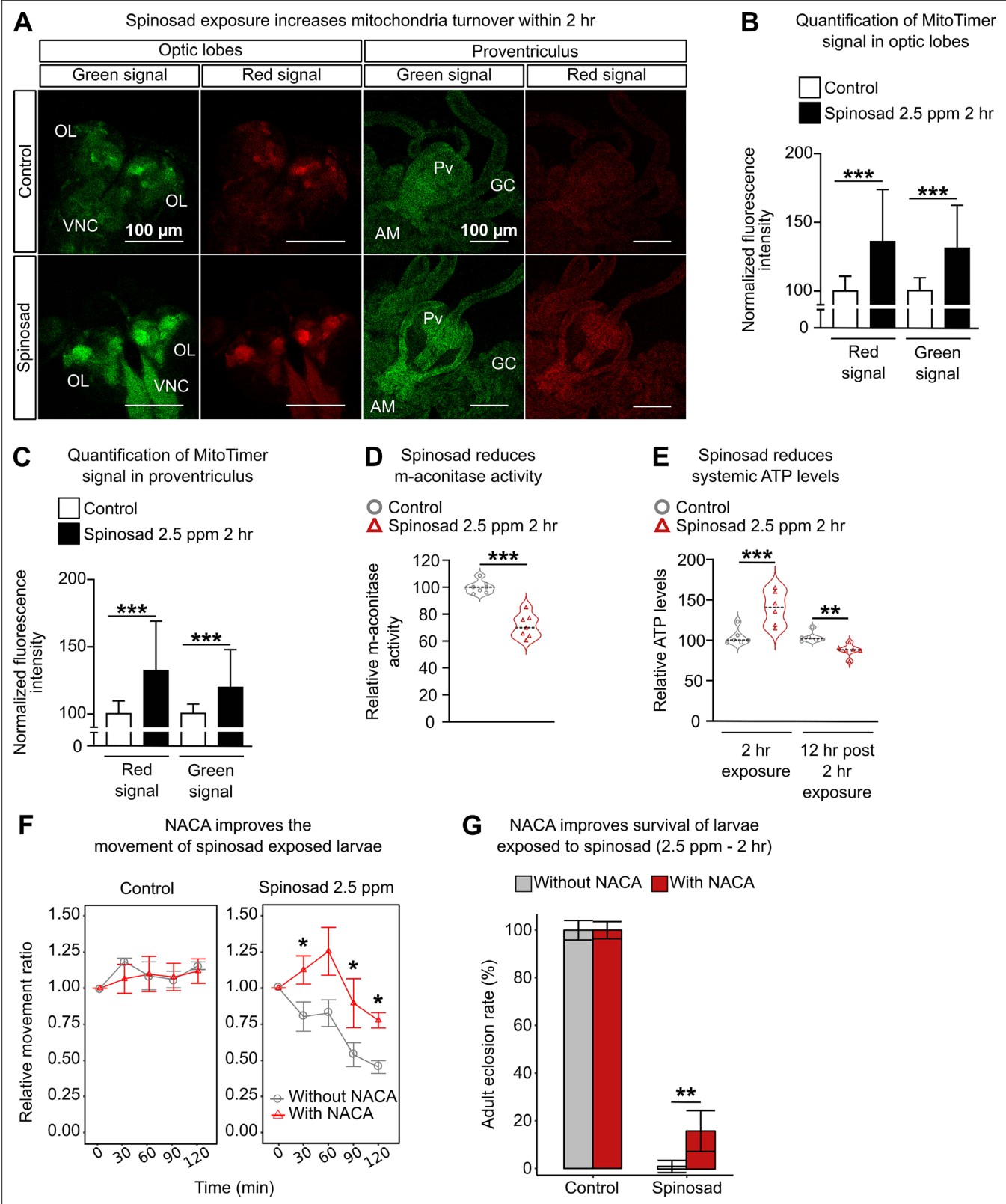

**Figure 3.** Spinosad exposure impacts mitochondria and energy metabolism, and antioxidant treatment reduces spinosad toxicity. (**A**) Optic lobes of the brain and proventriculus of MitoTimer reporter expressing larvae. 2.5 parts per million (ppm) spinosad exposure for 2 hr increased the signal of healthy (green) and unhealthy (red) mitochondria. (**B**) Normalized mean fluorescence intensity of MitoTimer signals in optic lobes (n = 20 larvae/treatment; three image sections/larva). (**C**) Normalized mean fluorescence intensity of MitoTimer signals in proventriculus (n = 20 larvae/treatment; three image sections/

*Figure 3 continued on next page*

Figure 3 continued

larva). (**D**) Relative m-aconitase activity in whole Line14 larvae exposed to 2.5 ppm spinosad for 2 hr (n = 25 larvae/replicate; six replicates/treatment). (**E**) Relative systemic ATP levels in Line14 larvae immediately after the 2 hr exposure to 2.5 ppm spinosad and 12 hr post 2 hr exposure (n = 20 larvae/ replicate; six replicates/ treatment). (**F**) Pretreatment with N-acetylcysteine amide (NACA) improves the movement of spinosad-exposed Line14 larvae. Dose–response to insecticide by an assay of larval movement over time, expressed in terms of relative movement ratio (n = 25 larvae/replicate; four replicates/treatment). (**G**) Pretreatment with NACA improves survival of Line14 larvae exposed to spinosad. Adult eclosion rate (%) (n = 100 larvae/ treatment). OL, optic lobe; VNC, ventral nerve cord; Pv, proventriculus; GC, gastric caeca; AM, anterior midgut. Error bars in (**B**) and (**C**) represent mean ± SD; in (**F**), mean ± SEM; and in (**G**), corrected percentage survival (Abbot's correction). Microscopy images were obtained with a Leica SP5 Laser Scanning Confocal Microscope. (**B, C, E**) One-way ANOVA followed by Tukey's honestly significant difference test; (**D, F, G**) Student's unpaired t-test. **p<0.01, ***p<0.001.

The online version of this article includes the following figure supplement(s) for figure 3:

**Figure supplement 1.** Spinosad exposure increases mitochondrial reactive oxygen species (ROS) signal.

genes, *Marf* and *ND42*, in *Drosophila* neurons increases the levels of ROS in the brain and precipitates the accumulation of LD in glial cells (*Liu et al., 2015*). *Martelli et al., 2020* showed that the knock-down of the same mitochondrial genes in *Drosophila* neurons can also precipitate the accumulation of LD in fat bodies and a reduction of LD in Malpighian tubules. Such changes in the lipid environment of metabolic tissues were recapitulated by low imidacloprid exposures, which, like spinosad, also causes an increase of ROS levels in the brain that further spreads to the anterior midgut (*Martelli et al., 2020*). To test if spinosad could also affect the lipid environment of metabolic tissues, LD numbers were assessed using Nile Red staining. Larvae exposed to 2.5 ppm spinosad for 2 hr showed a 52% increase in the area covered by LD in the fat body (*Figure 5A and B*), with a significant reduction in the number of large LD and an increase in small LD (*Figure 5—figure supplement 1*).

Once inside the insect body, spinosad could theoretically access any tissue via the open circulatory system. Given that Dα6 is present in the nervous system (*Perry et al., 2015*; *Somers et al., 2015*), and that elevated levels of ROS were observed earlier in the brain than in metabolic tissues, we addressed the following question: Do the interactions between spinosad and Dα6 in the brain provide the signal that ultimately leads to the observed disturbance of the lipid environment in the metabolic tissues? No expression of *Dα6* has been reported in gut and fat body, but it is abundantly and widely expressed in most CNS neurons with little to no expression in glia (*Li et al., 2021*). Two different *Dα6* loss-of-function mutants (Line14 *Dα6^{nx}* loss-of-function mutant and Canton-S *nAChRα6 knockout*) and their respective genetic background control lines were tested for LD. Larvae were submitted to the same exposure conditions, 2.5 ppm of spinosad for 2 hr. Neither of the mutants tested showed an increase in the area occupied by LD, compared to their respective genetic background, under conditions of spinosad exposure (*Figure 5A and B*). Interestingly, Line14 *Dα6^{nx}* loss-of-function mutant and Canton-S *Dα6 KO* mutant showed on average 16 and 20% larger area covered by LD in fat body than their respective background control lines (*Figure 5A and B*). These data show that the *Dα6* loss of function by itself affects the lipid environment of metabolic tissues.

Treatment with NACA prior to insecticide exposure significantly ameliorated spinosad effects on fat body LD accumulation (*Figure 5C and D*). This indicates that ROS induced by spinosad exposure is indeed involved in the LD phenotype in fat bodies. However, measurements of ROS levels in fat bodies showed no differences between exposed and unexposed larvae (*Figure 5E and F*). This result indicates that the presence of a ROS signal other than the one measured here causes the bulk redistribution of lipids into LD. That no accumulation of LD was observed in the absence of *Dα6* and presence of spinosad (*Figure 5A and B*) suggests that in wild-type flies spinosad exposure generates a signal in the brain that triggers fat body to respond. It is possible that oxidizing agents, such as peroxidized lipids, are transported through the hemolymph to the fat body (*Martelli et al., 2020*; *Padmanabha and Baker, 2014*). Alternatively, other secreted signals from the brain affected fat body metabolism.

To test for alterations of lipid levels in hemolymph, we used the vanillin assay (*Cheng et al., 2011*). Whereas the wild-type Line14 and Canton-S strains showed an average increase of 14 and 11% in lipid levels in hemolymph in response to spinosad, neither of the *Dα6* loss-of-function mutants showed significant changes after exposure (*Figure 5G*). This result supports the notion that changes in the lipid environment upon spinosad exposure depend on the insecticide interacting with *Dα6* receptors. However, *Dα6* mutants showed higher lipid levels than their respective wild-type controls (*Figure 5G*).

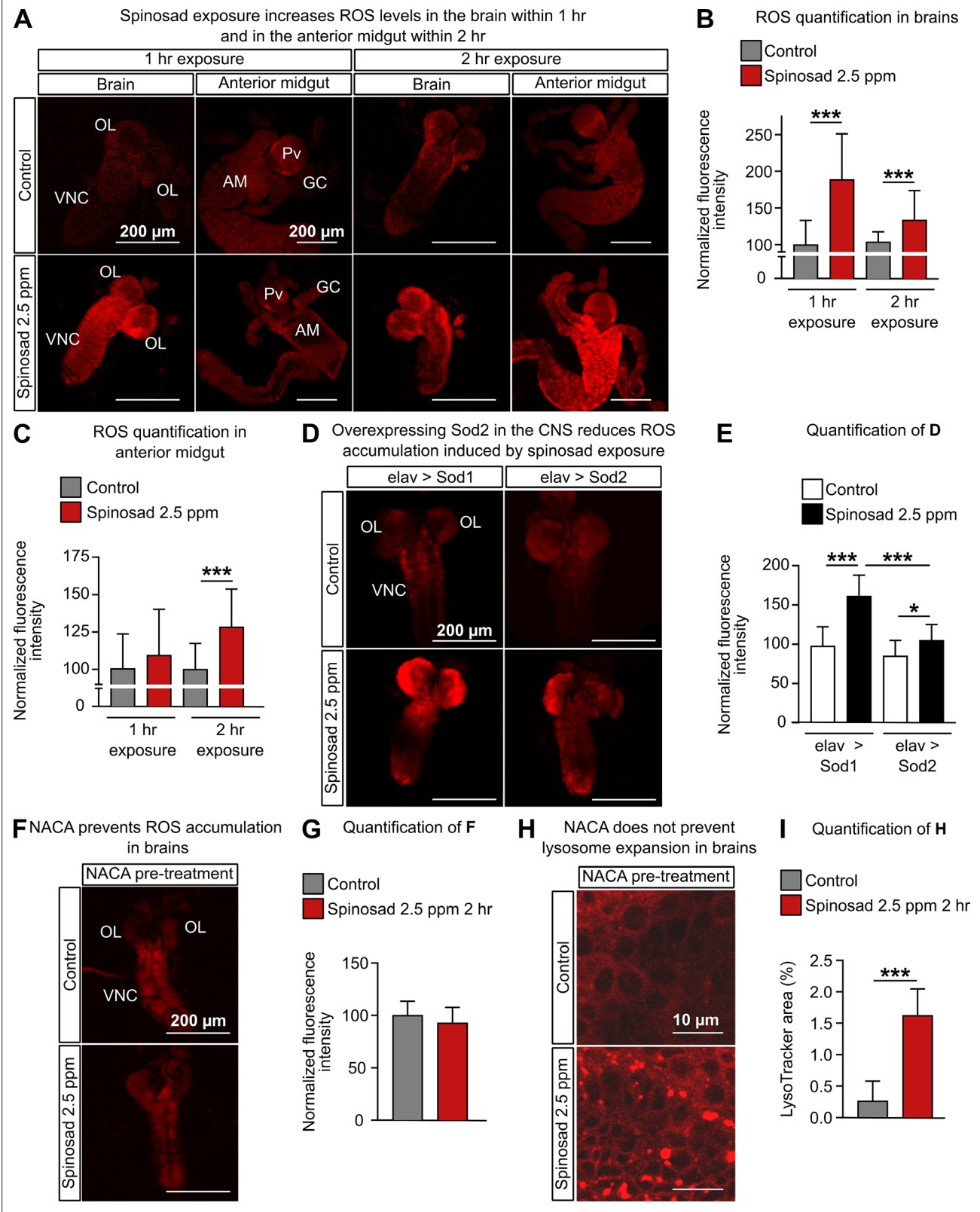

**Figure 4.** Spinosad exposure increases oxidative stress, and antioxidants prevent reactive oxygen species (ROS) accumulation, but not lysosome expansion. (**A**) Dihydroethidium (DHE) staining of ROS levels in the brain and anterior midgut of Line14 larvae exposed to 2.5 parts per million (ppm) spinosad for either 1 hr or 2 hr. (**B**) DHE normalized fluorescence intensity in brains (n = 15 larvae/treatment; three sections/larva). (**C**) DHE normalized fluorescence intensity in anterior midgut (n = 15 larvae/treatment; three sections/larva). (**D**) DHE staining of ROS levels in the brains of larvae expressing

*Figure 4 continued on next page*

*Figure 4 continued*

Sod2 (elav-Gal4>UAS-Sod2) or Sod1 (control cross; elav-Gal4>UAS-Sod1) in the central nervous system and exposed to 2.5 ppm spinosad for 2 hr. (**E**) DHE normalized fluorescence intensity in brains (n = 7 larvae/genotype/treatment; three sections/larva). (**F**) DHE staining of ROS levels in the brain of Line14 larvae treated with N-acetylcysteine amide (NACA) and exposed to 2.5 ppm spinosad for 2 hr. (**G**) DHE normalized fluorescence intensity in brains (n = 8 larvae/treatment; three sections/larva). (**H**) LysoTracker staining showing lysosomes in brains of Line14 larvae treated with NACA and exposed to 2.5 ppm spinosad for 2 hr. (**I**) LysoTracker area (%) (n = 8 larvae/treatment; three optic lobe sections/larva). OL, optic lobe; VNC, ventral nerve cord; Pv, proventriculus; GC, gastric caeca; AM, anterior midgut. Error bars represent mean ± SD. Microscopy images were obtained with a Leica SP5 Laser Scanning Confocal Microscope. (**B, C, E**) One-way ANOVA followed by Tukey's honestly significant difference test; (**G, I**) Student's unpaired *t*-test. *$p < 0.05$, ***$p < 0.001$.

The online version of this article includes the following figure supplement(s) for figure 4:

**Figure supplement 1.** *Dα6 KO* mutants show no increase in ROS levels after spinosad exposure.

Spinosad doses that do not impact larval survival were also examined for perturbations in the lipid environment. Doses of 0.5 ppm for 2 hr or 0.1 ppm for 4 hr were used as they had no impact on survival rate (*Figure 5—figure supplement 2*). Both doses caused on average a 29% increase in the area occupied by LD in fat bodies (*Figure 5—figure supplement 2*). This impact is smaller than that observed with 2.5 ppm, indicating that this phenotype is dose dependent. Once again, an increase in the number of small LD and reduction in the number of large LD was observed (*Figure 5—figure supplement 2*). In contrast, both doses caused on average a 72% reduction in the total number of LD in the Malpighian tubules (*Figure 5—figure supplement 2*).

## Spinosad triggers major alterations in the lipidome pointing to impaired cell membrane function and a severe decrease in mitochondrial cardiolipins

To further investigate the impacts on lipid environment, we performed a lipidomic analysis on whole larvae exposed to 2.5 ppm spinosad for 2 hr. Significant changes were observed in the levels of 88 lipids out of the 378 detected by mass spectrometry (*Figure 6A*, *Figure 6—source data 1*). A significant portion of the changes in lipids corresponds to a reduction in phosphatidylcholine (PC), phosphatidylethanolamine (PE), and some triacylglycerol (TAG) species. Multivariate analysis (*Figure 6B*) indicates that the overall lipidomic profiles of exposed larvae form a tight cluster that is distinct from the undosed control. The use of whole larvae for lipidomic analysis reduces the capacity to detect significant shifts in lipid levels that predominantly occur in individual tissues but allows the identification of broader impacts on larval biology. In this context, the observed 65% reduction in the levels of identified cardiolipins (CLs) is particularly noteworthy (*Figure 6C*). CLs are highly enriched in mitochondria and are required for the proper function of the electron transport chain, especially Complex 1, the major ROS generator when dysfunctional (*Quintana et al., 2010*; *Ren et al., 2014*), consistent with the increase in ROS in mitochondria described earlier.

## Chronic low exposure to spinosad causes neurodegeneration and progressive loss of vision

Next, we investigated the effects of chronic exposure of low levels of spinosad in adult female virgin flies. A dose of 0.2 ppm spinosad, which kills 50% of adults within 25 days (*Figure 7A*), was used in all chronic exposure experiments. Two different behavioral outputs were initially assessed: bang sensitivity (BS) and climbing. BS is associated with seizures in flies. Several fly mutants that exhibit BS have been previously described (*Kanca et al., 2019*; *Saras and Tanouye, 2016*). The assay measures the time it takes for flies to recover to a standing position following mechanical shock induced by vortexing. Wild-type flies recover in a few seconds, whereas BS mutants require typically between 5 and 60 s. Exposures for 10 and 20 days to 0.2 ppm increased the BS phenotype that has been associated with perturbations in synaptic transmission (*Saras and Tanouye, 2016*). These can arise from various defects including defective channel localization, neuronal wiring, and mitochondrial metabolism (*Fergestad et al., 2006*; *Figure 7B*). Exposed flies also performed poorly in climbing assays, a phenotype that is often linked to neurodegeneration (*McGurk et al., 2015*). Indeed, 16, 73, and 84% of flies failed to climb after 1, 10, and 20 days of exposure, respectively (*Figure 7C*). These data suggest that low doses of spinosad induce neurodegenerative phenotypes.

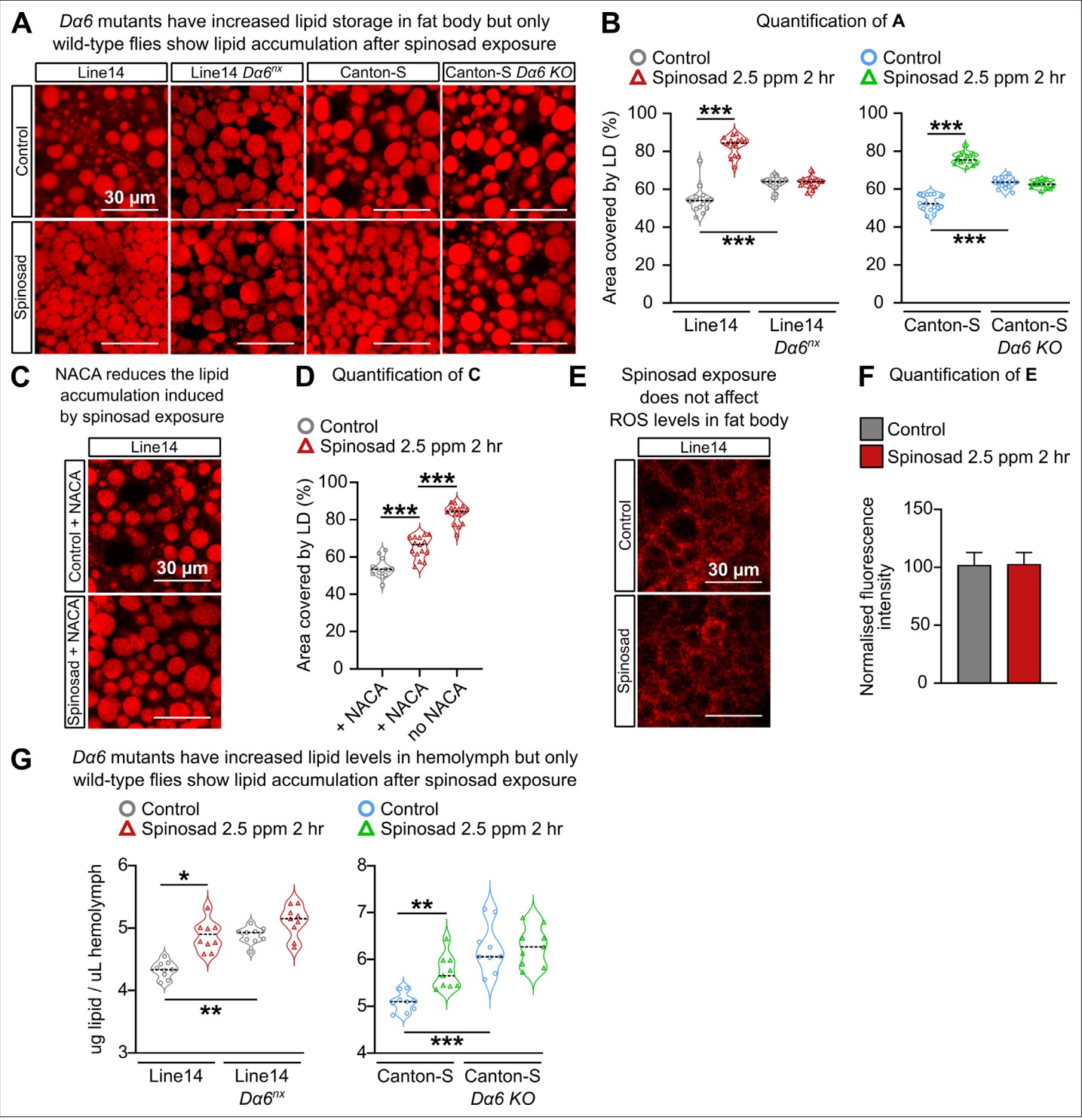

**Figure 5.** Spinosad triggers reactive oxygen species (ROS)-driven lipid changes in metabolic tissues of wild-type larvae but not *Dα6* loss-of-function larvae. (**A**) Nile Red staining showing lipid droplets in larval fat bodies of Line14 and Canton-S strains and their respective *Dα6* loss-of-function mutant strains. Larvae exposed to 2.5 parts per million (ppm) spinosad for 2 hr. (**B**) Area covered by lipid droplets in fat body (%) (n = 3 larvae/treatment/genotype; five image sections/larva). (**C**) Nile Red staining showing lipid droplets in fat bodies of Line14 larvae treated with N-acetylcysteine amide (NACA) and exposed to 2.5 ppm spinosad for 2 hr. (**D**) Area covered by lipid droplets in fat body (%) (n = 3 larvae/treatment; five image sections/larva). (**E**) Dihydroethidium (DHE) staining of ROS levels in the fat body of Line14 larvae exposed to 2.5 ppm spinosad for 2 hr. (**F**) DHE normalized fluorescence intensity in fat body (n = 3 larvae/treatment; five sections/larva). (**G**) Amount of lipids in hemolymph (μg/μL) by colorimetric vanillin assay of Line14 and Canton-S larvae and their respective *Dα6* loss-of-function mutants. Larvae exposed to 2.5 ppm spinosad for 2 hr (n = 10 replicates/treatment/time point;

*Figure 5 continued on next page*

*Figure 5 continued*

30 larvae/replicate). Microscopy images were obtained with a Leica SP5 Laser Scanning Confocal Microscope. Error bars in (**F**) represent mean ± SD. (**B**, **D**, **G**) One-way ANOVA followed by Tukey's honestly significant difference test; (**F**) Student's unpaired *t*-test. *p<0.05, **p<0.01, ***p<0.001.

The online version of this article includes the following figure supplement(s) for figure 5:

**Figure supplement 1.** Impact of spinosad exposure on lipid droplets dynamics in fat body.

**Figure supplement 2.** Spinosad doses that do not affect survival impact the larval lipid environment.

The same phenotypes were also assessed in adult female virgin *Dα6 KO* mutants. Unexposed mutant flies show a significant reduction in longevity compared to unexposed wild-type individuals, but that difference is only noticeable later in life; Canton-S *Dα6 KO* mutants have a median survival of 68 days compared to 82 for Canton-S (***Figure 7D***). A 25-day exposure to 0.2 ppm spinosad leads to a 91% survival of Canton-S *Dα6 KO* mutants, whereas only 40% of Canton-S wild-type flies survive this exposure (***Figure 7E***). No changes in BS or climbing ability were observed between exposed and unexposed *Dα6 KO* mutants over the course of a 20-day exposure (***Figure 7F and G***). However, at the 20-day time point, twice as many of the unexposed *Dα6 KO* mutants failed to climb (36%) compared to unexposed Canton-S wild-type flies (18%). (***Figure 7G***). These data support that the deleterious effects of spinosad are mediated by its binding to *Dα6* receptors. They also indicate that loss of *Dα6* by itself causes mild but significant behavioral and life span phenotypes not previously reported.

We then examined the retinas of adult flies for evidence of neurodegenerative markers, such as the accumulation of LD based on Nile Red staining (***Liu et al., 2015***). Adult virgin females (3–4-day-old) exposed to 5 ppm of spinosad for 6 hr showed a significant accumulation of LD in the glial cells of the retina (***Figure 8A and B***), indicating the ability of spinosad to induce another ROS-related phenotype (***Liu et al., 2017***) within a few hours of exposure. Chronic exposures to 0.2 ppm, however, did not lead to a clear LD phenotype in glial cells of retinas. However, Nile Red-positive accumulations were observed decorating the plasma membrane of photoreceptor neurons (PR) after 10 and 20 days of exposure (***Figure 8C and D***). Even though *Dα6* is not expressed in the retina, it is widely expressed in the adult brain, notably in the lamina neurons that synapse with the PR (***Figure 9A***). The accumulation of lipids in neurons suggests that the postsynaptic cells that express *Dα6* somehow affect lipid production or transfer to PR.

The retinas of unexposed *Dα6 KO* mutants were also examined. Adult virgin females, 1, 10, and 20 days old, showed no Nile Red-positive accumulations in retinas. However, 37% of 10-day-old mutants and 50% of 20-day-old mutants showed abnormal rhabdomeres (***Figure 8E and F***). These visual system defects have not been described previously in a *Dα6* KO mutant and are obviously different from the ones arising from the interaction between *Dα6* and spinosad.

Given the Nile Red-positive accumulation in retinas of chronically exposed flies, we investigated the impact on visual function. Electroretinograms (ERGs) were performed at regular intervals over the 20 days of exposure (***Figure 9B and C***). ERG recordings measure the impact of light pulses on PR. The on-transient is indicative of synaptic transmission between PR and postsynaptic cells, whilst the amplitude measures the ability of photons to impact the phototransduction cascade (***Wang and Montell, 2007***). A large reduction in the on-transient was observed from day 1 of exposure, whereas the amplitude was only significantly impacted after 20 days of exposure (***Figure 9B and C***). The reduction in the on-transient is evidence of a rapid loss of synaptic transmission in laminar neurons (***Wang and Montell, 2007***) and hence impaired vision after just 1 day of exposure. In examining the visual system of *Dα6 KO* mutants reared without spinosad, mild impacts were identified in ERG amplitude but a very significant reduction in on-transient was also observed, consistent with a requirement for *Dα6* in lamina postsynaptic cells of the PR (***Figure 9D and E***).

To investigate the ultrastructure of the PR synapses, we used transmission electron microscopy. Compared to unexposed flies (***Figure 10A***), severe morphological alterations were detected in transverse sections of the lamina of flies exposed for 20 days (***Figure 10B–F***). Vacuoles in photoreceptor terminals or postsynaptic terminals of lamina neurons were observed in the lamina cartridges (***Figure 10B***). On average, 75% of the lamina cartridges contained vacuoles (***Figure 10E***). Large intracellular compartments were also observed in dendrites of the postsynaptic neurons in the lamina (***Figure 10C and D***). These do not correspond to normal structures found in healthy lamina (***Figure 10A***) and suggest the presence of aggregated material. The lamina of exposed flies also

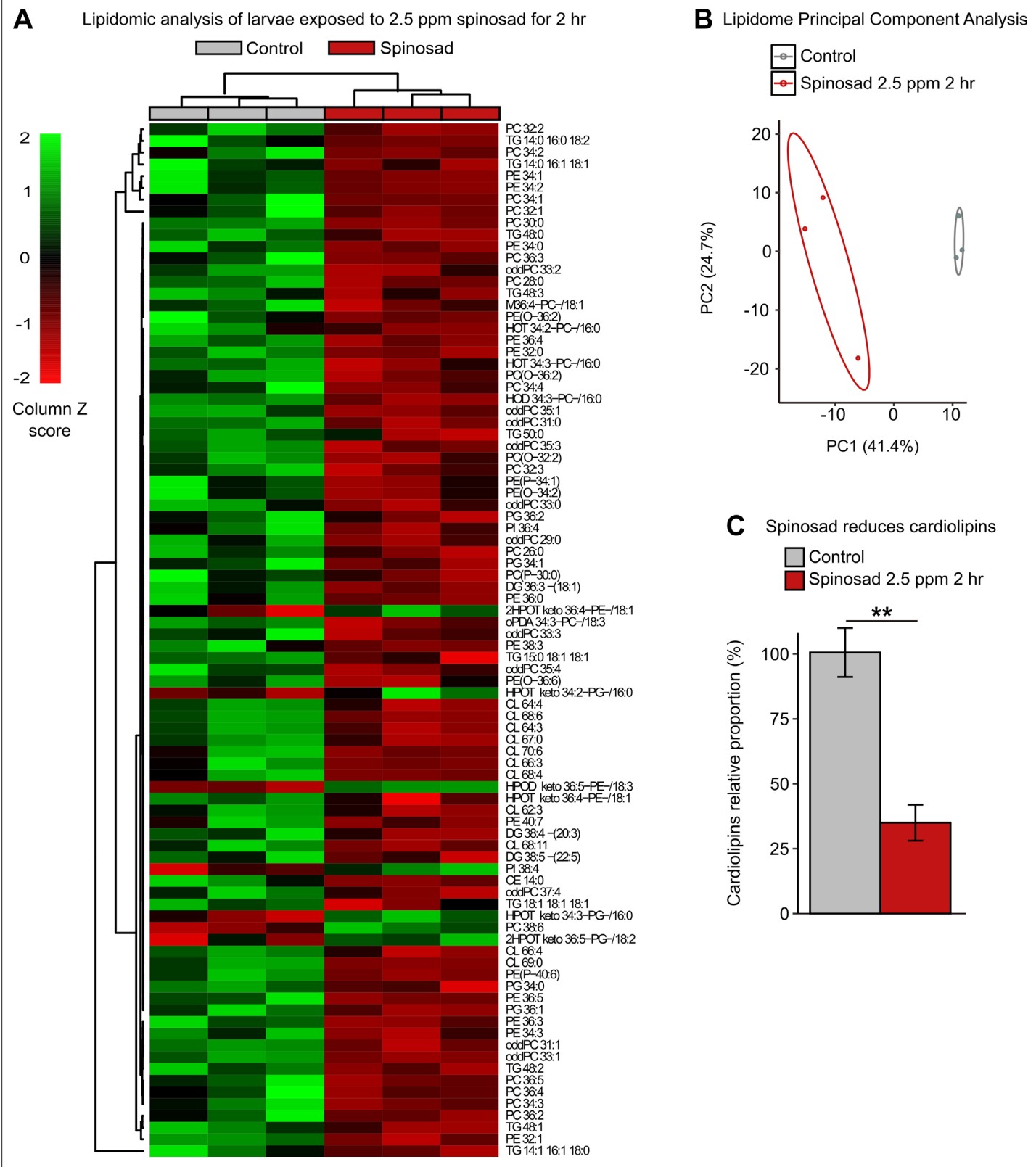

**Figure 6.** Spinosad disturbs the lipid profile of exposed larvae. Lipidomic profile of larvae exposed to 2.5 parts per million (ppm) spinosad for 2 hr (n = 10 larvae/replicate; three replicates/treatment). (**A**) 88 lipid species out of the 378 identified were significantly affected by insecticide treatment (one-way ANOVA, Turkey's honestly significant difference test, p<0.05). The cell colors represent the z-scores, that is, the standardized scores on the same scale, calculated by dividing a score's deviation by the standard deviation in the row. The features are color-coded by row with red indicating low intensity and

*Figure 6 continued on next page*

*Figure 6 continued*

green indicating high intensity. (**B**) Principal component analysis of 378 lipid species. Each dot represents the lipidome data sum of each sample. First component explains 41.4% of variance and second component explains 24.7% of variance. (**C**) Relative proportion of cardiolipins in exposed animals versus control. Error bars represent mean ± SEM, Student's unpaired *t*-test, \*\*p<0.01.

The online version of this article includes the following source data for figure 6:

**Source data 1.** Impact of spinosad on the lipidomic profile.

showed a mean 34% increase in the number of mitochondria (*Figure 10F*), many of which appear defective (*Figure 10B*). No morphological alterations were detected by TEM in lamina of 20-day-old *Dα6 KO* mutants reared in the absence of spinosad (*Figure 10G and H*).

Lastly, hematoxylin and eosin stain (H&E) of adult flies painted a broader picture of the neurodegeneration caused outside the visual system by chronic low-dose exposure to spinosad. 20 days of exposure caused numerous vacuoles in the central brain (*Figure 10I*). On average, 17% of the total central brain area was consumed by vacuoles in exposed flies (*Figure 10J*).

## Discussion

### Spinosad antagonizes neuronal activity

In this study, we explore the mechanisms and consequences of exposure to low doses of spinosad upon binding to its target *Dα6*, comparing these phenotypes side by side with the ones caused by *Dα6* loss of function. Low levels of spinosad lead to a lysosomal dysfunction associated with mitochondrial stress, elevated levels of ROS, lipid mobilization defects, and severe neurodegeneration. Spinosad has been characterized as an allosteric modulator of the activity of its primary target, the Dα6 subunit, causing fast neuron over-excitation (*Salgado, 1998*). Here, the capacity of spinosad to interact with its target to stimulate the flux of $Ca^{2+}$ into neurons was quantified. The results obtained with the GCaMP assay showed that spinosad caused no detectable increase or decrease in $Ca^{2+}$ flux into *Dα6*-expressing neurons, but reduces the cholinergic response (*Figure 1D and E*). Given that spinosad binds to the C terminal region of the protein (*Crouse et al., 2018*; *Puinean et al., 2013*; *Somers et al., 2015*), these findings are consistent with a noncompetitive antagonist mode of action for spinosad on nAChRs.

### Spinosad's toxicity involves more than causing an absence of Dα6 from neuronal membranes

*Dα6* loss-of-function mutants are viable and highly resistant to spinosad (*Figure 1C*; *Perry et al., 2007*), yet the loss of Dα6 from the membrane precipitated by exposure to spinosad in wild-type flies leads to death. This creates a conundrum that can be resolved if a significant component of spinosad's toxicity is due to molecular events that play out elsewhere in the cell. Blocked neuronal receptors can be recycled from the plasma membrane through endocytosis (*Saheki and De Camilli, 2012*). Here, we demonstrate that Dα6 is removed from neuronal membranes in response to spinosad exposure (*Figure 2A*) and localizes to lysosomes (*Figure 2G*), resulting in a significant lysosomal expansion (*Figure 2C*) and increase in ROS levels (*Figure 4A and B*). In exposed *Dα6* mutants, spinosad does not lead to lysosome expansion (*Figure 2D*) or an increase in ROS levels in the brain (*Figure 4—figure supplement 1*). These two phenotypes precede all other phenotypes observed in wild-type exposed larvae. In unexposed mutants, the mild ROS levels found in brains (*Figure 4—figure supplement 1*) seem to be a direct consequence of Dα6 absence. Indeed, *Dα6* has been associated with the response to oxidative stress and *Dα6* mutants are more susceptible to oxidative damage (*Weber et al., 2012*). The mildly elevated ROS levels in unexposed *Dα6* mutants cannot be ignored, nor can the altered lipid environment (*Figure 5A, B, and G*), a minor reduction in longevity and increased climbing defects with age (*Figure 7D and G*), rhabdomere degeneration (*Figure 8E and F*), as well as loss of synaptic transmission in photoreceptors (*Figure 9D and E*). These are all previously unreported metabolic and nervous system defects associated with *Dα6* loss of function.

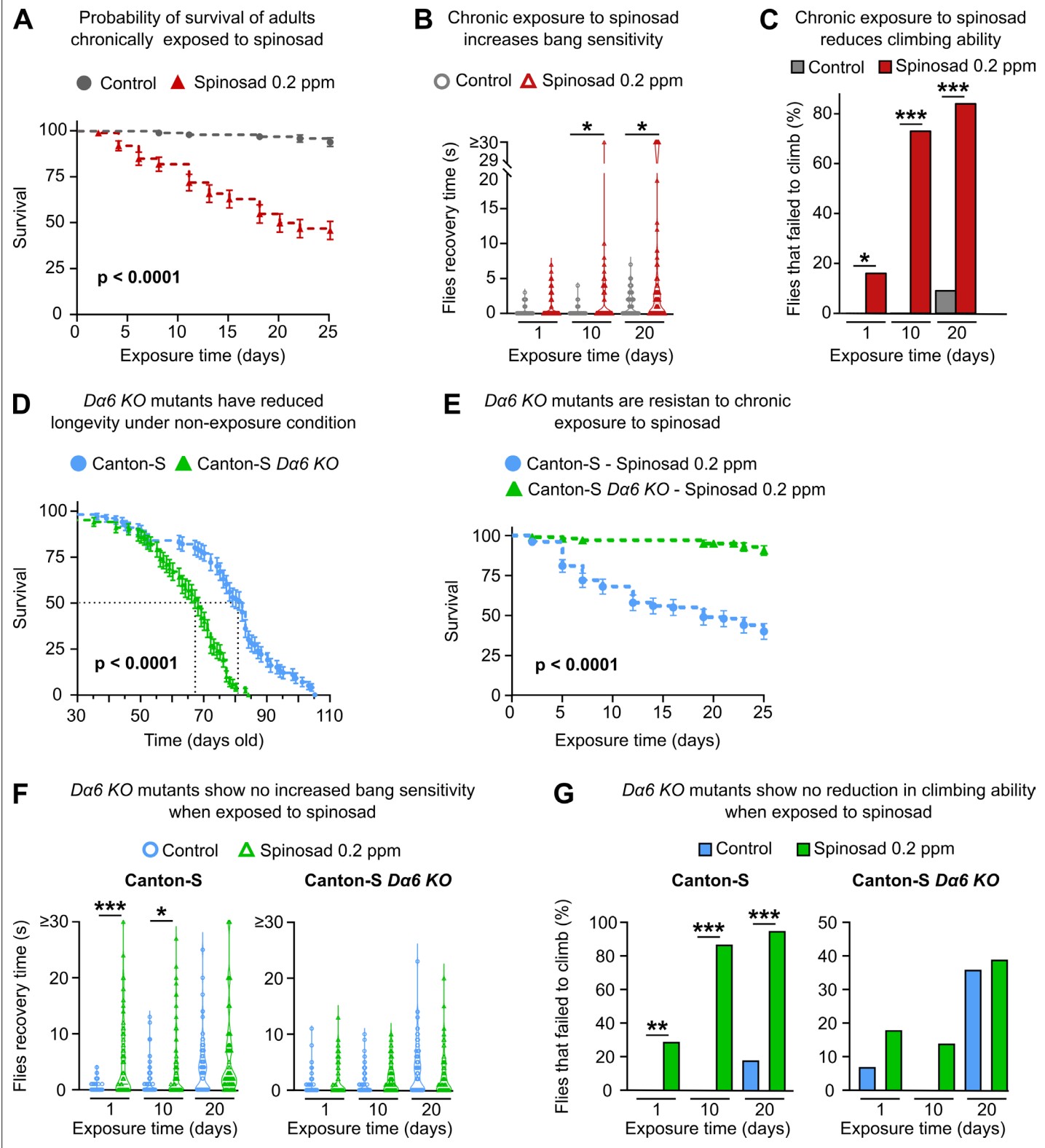

**Figure 7.** Chronic effects of spinosad exposure are more severe than loss of *Dα6* expression in adult virgin females. (**A**) A chronic exposure to 0.2 parts per million (ppm) spinosad kills 50% of flies within 25 days (n = 25 flies/replicate; four replicates/treatment). (**B**) Chronic exposure to 0.2 ppm spinosad increases bang sensitivity after 10 and 20 days of exposure. Time to regain normal standing posture (seconds) after flies were vortexed in a clear vial for 10 seconds (n = 100 flies/time point/treatment). (**C**) Chronic exposure to 0.2 ppm spinosad reduces climbing ability. Percentage of flies that failed to climb after 1, 10, and 20 days of exposure (n = 100 flies/time point/treatment). (**D**) Longevity of unexposed Canton-S and Canton-S Dα6 KO mutants (n =

*Figure 7 continued on next page*

*Figure 7 continued*

100 flies/genotype). (**E**) Chronic exposure to 0.2 ppm spinosad for 25 days has no impact on survival of Canton-S Dα6 KO mutants (n = 25 flies/replicate; four replicates/genotype). (**F**) Chronic exposure to 0.2 ppm spinosad does not increase bang sensitivity of Canton-S Dα6 KO mutants. Time to regain normal standing posture (seconds) after flies were vortexed in a clear vial for 10 s (n = 100 flies/time point/genotype/treatment). (**F**) Chronic exposure to 0.2 ppm spinosad does not reduce climbing ability of Canton-S Dα6 KO mutants. Percentage of flies that failed to climb (n = 100 flies/time point/genotype/treatment). Error bars in (**A**), (**D**), and (**E**) represent mean ± SEM. (**A, D**, **E**) Kaplan–Meier method and the log-rank Mantel–Cox test. (**B, C, F, G**) Kruskal–Wallis followed by Dunn's multiple comparisons test. *p<0.05, ***p<0.001.

## Spinosad causes lysosomal dysfunction and mitochondrial stress

Lysosomal dysfunction and mitochondrial stress are the key players in the cascade of impacts following spinosad exposure. Whether spinosad molecules are ferried to lysosomes along with Dα6 subunits and accumulate into these organelles remains to be clarified. However, that the increased severity in the lysosomal phenotype after exposure ceases (*Figure 2C and E*) is consistent with the poisoning of these organelles. Lysosomes become enlarged as they accumulate undigested material, which typically lead to recycling problems (*Darios and Stevanin, 2020*). If spinosad remains bound to the receptor and is ferried into the lysosomes, it may contribute to a lysosomal dysfunction akin to lysosomal storage disease (LSD) (*Darios and Stevanin, 2020*). To date, there is little published evidence of spinosad metabolites in insects. Spinosyns are polyketide macrolactones, and we speculate that their complex molecular structure may prevent them from being degraded efficiently by metabolic enzymes in lysosomes, triggering a severe lysosomal dysfunction and expansion.

Extensive evidence connects LSDs with mitochondrial dysfunction (*Plotegher and Duchen, 2017*; *Stepien et al., 2020*; *Yambire et al., 2018*). Mitochondrial dysfunction is widespread in LSD and is involved in its pathophysiology, although the exact mechanisms remain unclear. Lysosomal disorders may lead to cytoplasmic accumulation of toxic macromolecules like ceramides (*Lin et al., 2018*), impaired mitophagy and dysregulation of intracellular $Ca^{2+}$ homeostasis, resulting in an increase in ROS generation and reduced ATP levels (*Plotegher and Duchen, 2017*). The severe lysosomal dysfunction observed here is a likely cause for mitochondrial dysfunction and increased ROS generation triggered by spinosad exposure. Treatment with antioxidant NACA was able to prevent the increase in ROS levels at the 2 hr time point exposure (*Figure 4F and G*) but did not prevent the lysosome expansion (*Figure 4H and I*). That lysosomes still expand in the absence of the ROS generated by mitochondria supports the notion that it precedes and triggers the mitochondrial stress (*Figure 3A–E*).

While we cannot rule out the generation of ROS by other mechanisms, we provide compelling evidence that a significant part of ROS that is generated by spinosad exposure is of mitochondrial origin, arguing that mitochondrial impairment is a key element of spinosad mode of action at low-dose exposure. The evidence is based on increased mitochondrial turnover and mito-roGFP 405/488 ratio, reduced activity of the ROS sensitive enzyme m-aconitase, and reduced ATP levels (*Figure 3*). In addition, we observed a highly significant reduction of CL levels (*Figure 6C*) typically associated with defects in the electron transport chain and increased ROS production as they are required for the anchoring of Complex 1 in mitochondria (*Dudek, 2017*; *Quintana et al., 2010*) Increased levels of ROS in the larval brain have been shown to disturb mitochondrial function triggering changes in lipid stores in metabolic tissues (*Martelli et al., 2020*). Oxidative stress promotes the redistribution of membrane lipids into LD, reducing their exposure to peroxidized lipids (*Bailey et al., 2015*; *Liu et al., 2015*). Here, we observed increases in lipid stores in the fat body (*Figure 5A and B*), reduction in LD numbers in the Malpighian tubules (*Figure 5—figure supplement 2*), and changes in lipid levels in the hemolymph (*Figure 5G*). Our lipidome analysis also revealed reduction of PE and PC levels (*Figure 6A*, *Figure 6—source data 1*), consistent with impaired membrane fluidity and altered LD dynamics (*Dawaliby et al., 2016*; *Guan et al., 2013*; *Krahmer et al., 2011*).

The use of the antioxidant NACA reduces the accumulation of LD in the fat body linking this phenotype to oxidative stress (*Figure 5C and D*). NACA also diminished spinosad toxicity by reducing the impact on larval movement and survival (*Figure 3F and G*). Exposure to spinosad (7.7 parts per billion for 24 hr) was previously shown to cause the vacuolation of epithelial cells of the midgut and Malpighian tubules of honeybees (*Apis mellifera*) (*Lopes et al., 2018*). It is not clear whether this is due to the spinosad:Dα6 interaction precipitating elevated levels of ROS. While the dysfunction of lysosomes and mitochondria and elevated levels of ROS can account for the defects observed here under conditions of spinosad exposure, we cannot rule out this insecticide having other impacts that

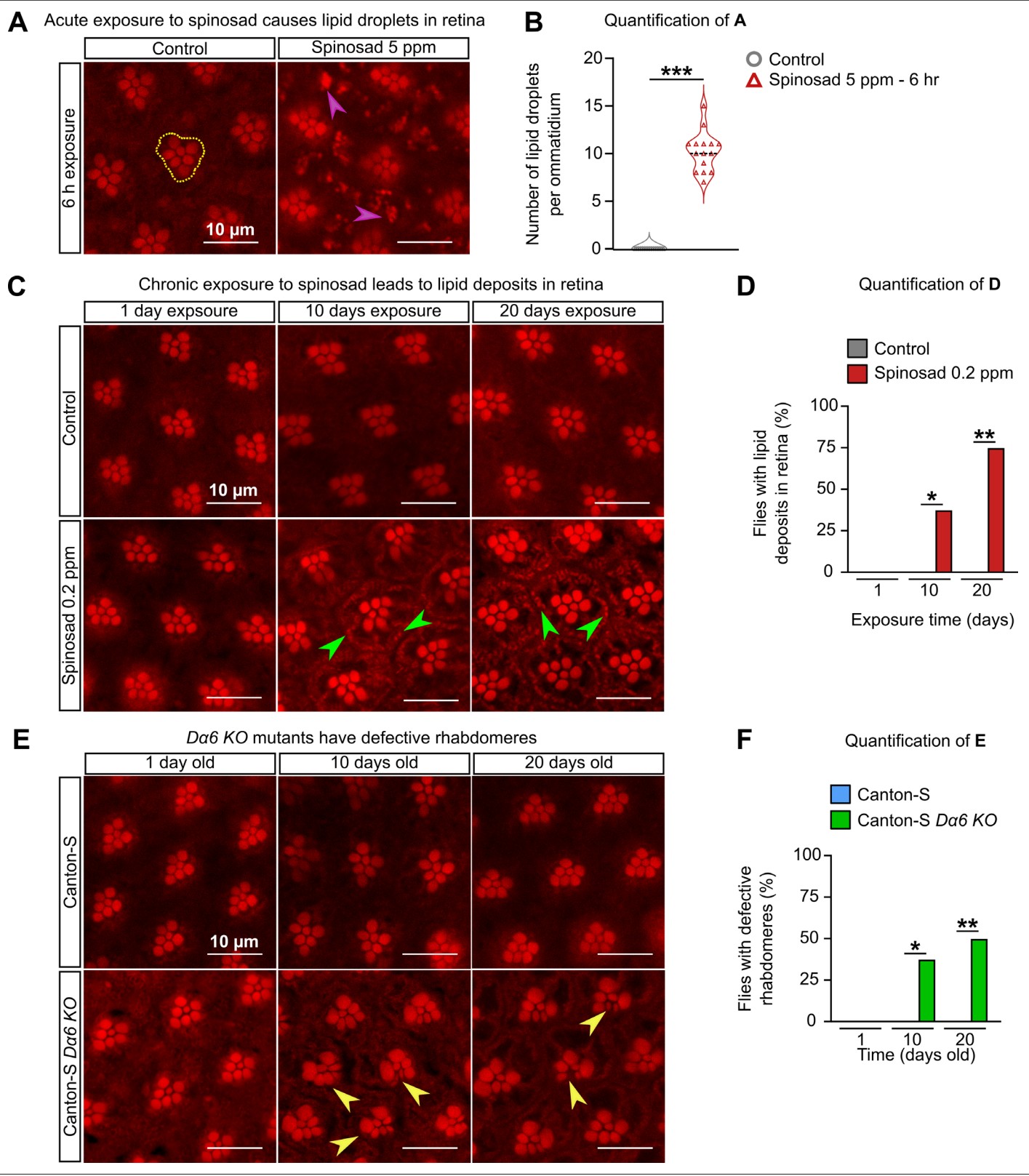

**Figure 8.** Chronic exposure to spinosad causes lipid deposits in retinas. (**A**) Nile Red staining of lipid droplets in the retinas of virgin females exposed to 5 parts per million (ppm) spinosad for 6 hr. Cluster of rhabdomeres delimited with yellow dotted line, purple arrowheads point to lipid droplets. (**B**) Number of lipid droplets per ommatidium (n = 5 flies/treatment; three image sections/retina). (**C**) Nile Red staining of lipid deposits in retinas of virgin females exposed to 0.2 ppm spinosad for 1, 10, and 20 days. Green arrowheads point to lipid deposits. (**D**) Percentage of flies with lipid deposits

*Figure 8 continued on next page*

*Figure 8 continued*

in retinas (n = 8 flies/time point/treatment). (**E**) Nile Red staining of defective rhabdomeres in retinas of virgin females Canton-S and Canton-S *Dα6 KO* mutants 1, 10, and 20 days old. Yellow arrowheads point to defective rhabdomeres. (**F**) Percentage of flies that show defective rhabdomeres (n = 8 flies/time point/genotype). Microscopy images were obtained with a Leica TCS SP8 Laser Scanning Confocal Microscope. (**B**) Student's unpaired *t*-test; (**D, F**) one-way ANOVA followed by Tukey's honestly significant difference test. *p<0.05, **p<0.01, ***p<0.001.

would contribute to its toxicity. Yet, the loss of *Dα6* strongly suppresses the phenotypes caused by exposure. Given that *Dα6* is not expressed in the gut and fat body, this suggests that the observed brain defects are at the root of most observed defects.

The LSD-like dysfunction is also likely the underlying cause for the severe vacuolation of adult central brain under spinosad chronic exposure (*Figure 10I and J*). Recycling defects in neuronal cells caused by LSD impair cell function, ultimately triggering neurodegeneration (*Darios and Stevanin, 2020*). A model for the low-dose mode of action of spinosad that is consistent with the data presented here is illustrated in *Figure 11*.

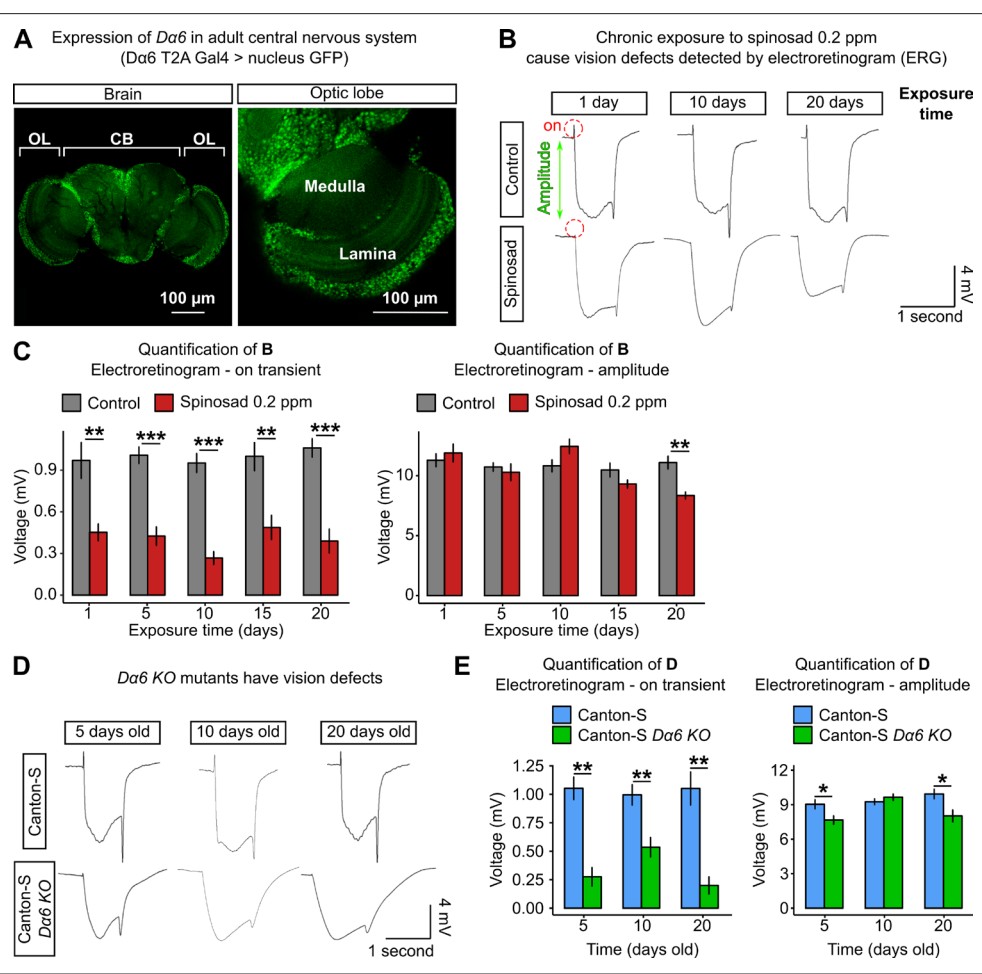

**Figure 9.** Chronic exposure to spinosad impairs the visual system. (**A**) Expression pattern of *Dα6* in the *Drosophila* adult female brain (*Dα6* T2A Gal4>UASGFP.nls). Detail of the expression in lamina and medulla (optic lobe). OL,- optic lobe; CB, central brain. (**B**) Electroretinograms (ERGs) of virgin females exposed to 0.2 parts per million (ppm) spinosad for 1, 10, and 20 days. Red dotted circles indicate the on-transient signal, and green arrow indicates the amplitude. (**C**) Measurements of on-transient signal and amplitude after 1, 5, 10, 15, and 20 days of exposure to 0.2 ppm spinosad (n = 8–10 flies/time point/treatment). (**D**) ERGs of virgin females Canton-S and Canton-S *Dα6 KO* mutants 5, 10, and 20 days old. (**E**) Measurements of on-transient signal and amplitude in Canton-S and Canton-S *Dα6 KO* mutants (n = 8–10 flies/time point/genotype). Microscopy images were obtained with a Leica TCS SP8 Laser Scanning Confocal Microscope. (**C**, **E**) One-way ANOVA followed by Tukey's honestly significant difference test. *p<0.05, **p<0.01, ***p<0.001.

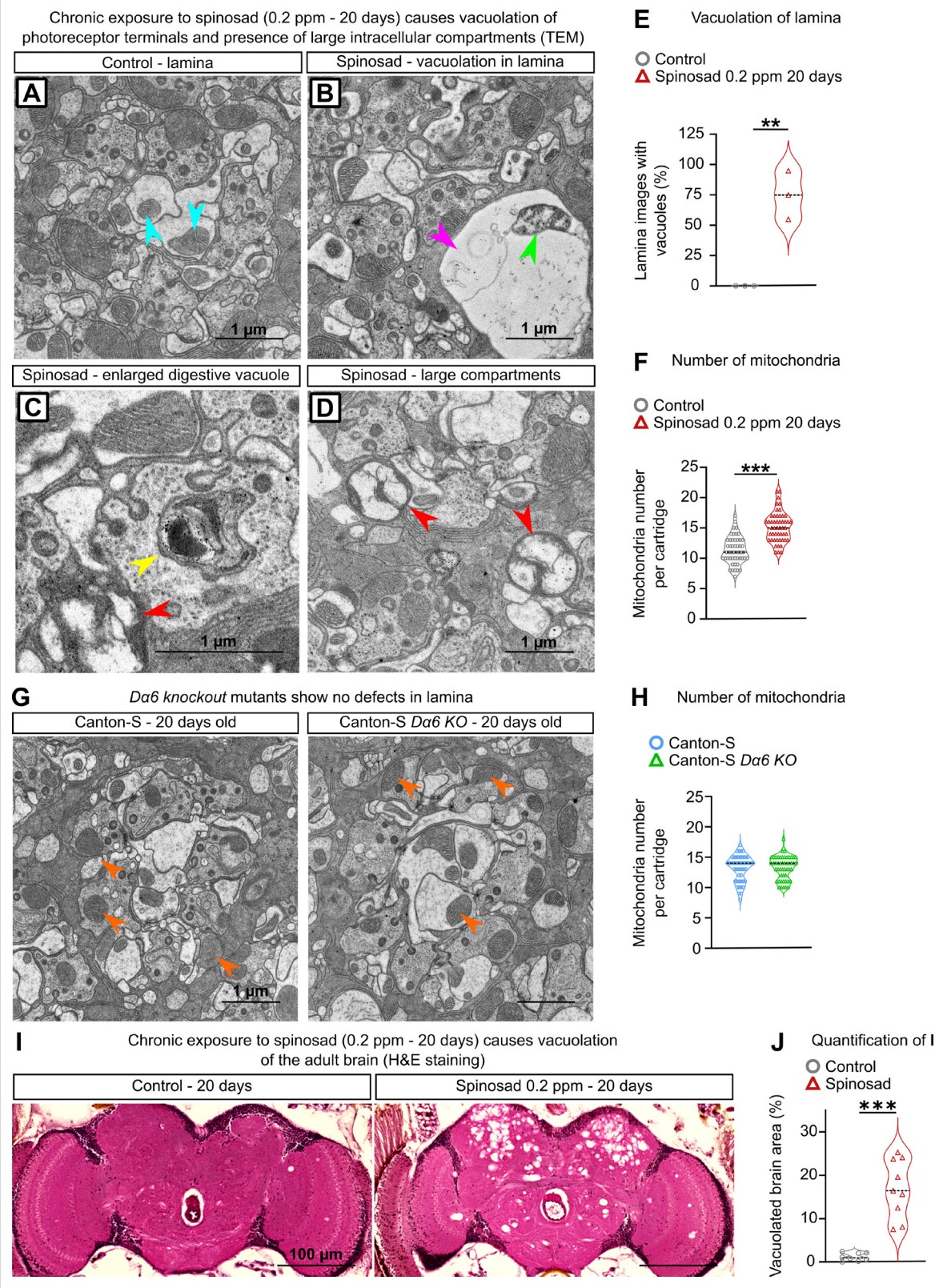

**Figure 10.** Chronic exposure to spinosad leads to neurodegeneration. (**A–D**) Transmission electron microscopy (TEM) of the lamina of virgin females exposed to 0.2 parts per million (ppm) spinosad for 20 days. (**A**) A regular cartridge of a control fly; blue arrowheads indicate normal mitochondria. (**B**) Cartridge of an exposed fly; pink arrowhead indicates a vacuole and green arrowhead indicates a defective mitochondrion. (**C, D**) Cartridges of exposed flies indicating an enlarged digestive vacuole (yellow arrowhead) and the presence of large intracellular compartments (red arrowheads).

*Figure 10 continued on next page*

Figure 10 continued

(**E**) Percentage of images showing vacuoles in lamina cartridges of exposed flies (10 images/fly; three flies/treatment). (**F**) Number of mitochondria per cartridge of exposed flies (n = 3 flies/treatment; 16 cartridges/fly). (**G**) TEM of the lamina of virgin 20-day-old females Canton-S and Canton-S *Dα6 KO* mutant. (**H**) Number of mitochondria per cartridge (n = 3 flies/genotype; 16 cartridges/fly). (**I**) Hematoxylin and eosin (H&E) staining of adult brain of virgin females exposed to 0.2 ppm spinosad for 20 days. (**J**) Quantification of neurodegeneration in terms of percentage of brain area vacuolated (n = 3 flies/treatment). (**E**, **F**, **H**, **J**) Student's unpaired *t*-test. **p<0.01, ***p<0.001.

## Spinosad causes neurodegeneration and affects behavior in adults

Both LSD (*Darios and Stevanin, 2020*) and oxidative stress (*Liu et al., 2017*; *Martelli et al., 2020*) can cause neurodegeneration. The evidence for spinosad-induced neurodegeneration comes from the reduced climbing ability and increased BS caused by chronic low-dose exposures (*Figure 7B and C*), vacuolation of the lamina cartridges, and severe vacuolation of the adult CNS (*Figure 10*). The neurodegeneration observed in the central brain (*Figure 10I and J*) seems to be largely contained to the functional regions of the optic tubercle, mushroom body, and superior lateral and medial proto- cerebrum. *Dα6* is widely expressed in the brain, including these regions (*Davie et al., 2018*; *Li et al., 2021*). These regions are important centers for vision and memory, and learning and cognition in flies (*Schürmann, 2016*). Neurodegeneration in these regions indicates that a wide range of behaviors will be compromised in exposed flies.

*Dα6* is not known to be expressed in PRs or glial cells, but its expression in lamina neurons (*Figure 9A*) supports its presence in postsynaptic cells of PR. The Nile Red-positive accumulations in PRs of wild-type flies after chronic spinosad exposure (*Figure 8C*) suggest the existence of cell nonau- tonomous mechanisms initiated by this insecticide in postsynaptic cells. *Liu et al., 2017* showed that ROS induce the formation of lipids in neurons that are transported to glia, where they form LD. Here, a ROS signal generated by spinosad exposure in postsynaptic cells might be carried to PRs, affecting lipid metabolism and triggering LD accumulation. This hypothesis needs further investigation.

## Rational control of insecticide usage

In the public domain, organic insecticides are often assumed to be safer than synthetic ones for the environment and nontarget insect species. The synthetic insecticide, imidacloprid, has faced intense scrutiny and bans because of its impact on the behavior of bees and the potential for this to contribute to the colony collapse phenomenon (*Wu-Smart and Spivak, 2016*). No other insecticide has been so comprehensively investigated, so it is not yet clear whether other chemicals pose similar risks.

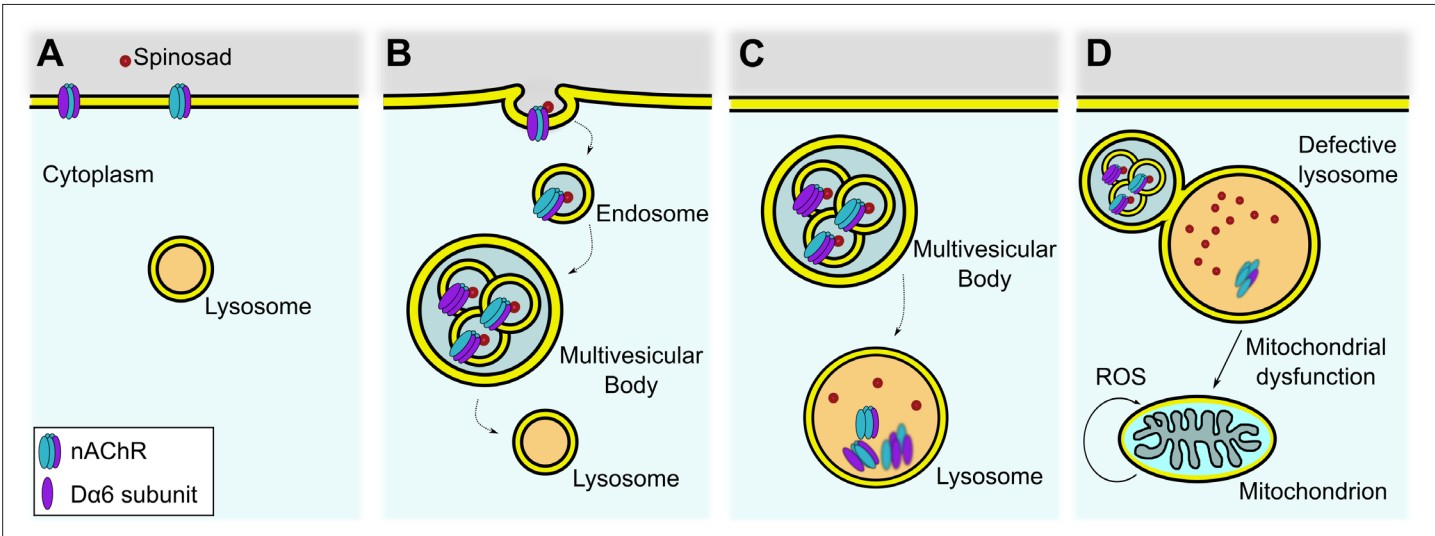

**Figure 11.** Proposed mechanism for internalization of spinosad after binding to Dα6 targets. (**A**) Spinosad binds to Dα6 subunit of nicotinic acetylcholine receptors (nAChRs) in the neuronal cell membranes. (**B**) The binding of spinosad leads to Dα6-containing nAChR blockage, endocytosis, and trafficking to lysosome. (**C**) Spinosad accumulates in lysosomes, while receptors and other membrane components are digested. (**D**) Expanded lysosomes due to accumulation of undigested material do not function properly, leading to cellular defects that may include mitochondrial dysfunction, increased mitochondrial reactive oxygen species (ROS) production, and eventually cell vacuolation and neurodegeneration.

This study has revealed disturbing consequences of low doses of an organic insecticide, spinosad. Based on similar assays deployed here, imidacloprid had a lower negative impact on *Drosophila* than spinosad (*Martelli et al., 2020*). At the low acute dose used here (2.5 ppm for 2 hr), imidacloprid has no effect on larval survival, while spinosad exposure is lethal. Given differences in the molecular weight, spinosad has a greater biological impact at lower concentration. 2.5 ppm corresponds to 3.4 µM spinosad and 9.8 µM imidacloprid. 4 ppm imidacloprid causes blindness and neurodegeneration, but no brain vacuolation under conditions of chronic exposure (*Martelli et al., 2020*), whereas 0.2 ppm spinosad causes vision loss and widespread brain vacuolation. Loss of function of Dα6 caused by mutation or by chronic exposure to spinosad leads to vision loss. This suggests that a wider analysis of *Dα6* mutant phenotypes may point to other consequences of spinosad exposure not detected in this study. Given that the Dα6 subunit has been shown to be a highly conserved spinosad target across a wide range of insects (*Perry et al., 2015*), it is likely that low doses of this insecticide will have similar impacts on other species. The susceptibility of different species to insecticides varies, so the doses required may differ. The protocols used here will be useful in assessing the risk that spinosad poses to other beneficial insects. Given the extent to which spinosad affects lysosomes, mitochondrial function, lipid metabolism, and vision, this insecticide very likely compromises the capacity of insects to survive in natural environments when exposed to a variety of stresses, including some of those that are being linked to insect population declines (*Cardoso et al., 2020*; *Sánchez-Bayo and Wyckhuys, 2019*).

Two clocks are ticking. The global human population is increasing, and the amount of arable land available for food production is decreasing. Thus, the amount of food produced per hectare needs to increase. Our capacity to produce enough food has been underpinned by the use of insecticides. Approximately 600,000 tonnes of insecticides are used annually around the world (*Aizen et al., 2009*; *Klein et al., 2007*), but sublethal concentrations found in contaminated environments can affect behavior, fitness, and development of target and nontarget insects (*Müller, 2018*). Despite their distinct modes of action, spinosad and imidacloprid produce a similar spectrum of damage (*Martelli et al., 2020*). This similarity arises because both insecticides trigger oxidative stress in the brain, albeit via different mechanisms. Several other insecticide classes such as organochlorines, organophosphates, carbamates, and pyrethroids have all been shown to promote oxidative stress (*Balieira et al., 2018*; *Karami-Mohajeri and Abdollahi, 2011*; *Lukaszewicz-Hussain, 2010*; *Terhzaz et al., 2015*; *Wang et al., 2016*). Many insect populations are exposed to a continuously changing cocktail of insecticides (*Kerr, 2017*; *Tosi et al., 2018*), most of which are capable of producing ROS. The cumulative effect of these different insecticides could be significant. Our research clarifies the mode of action of spinosad, highlighting the perturbations and damage that occur downstream of the insecticide:receptor interaction. Other chemicals should not be assumed to be environmentally safe until their low-dose biological impacts have been examined in similar detail.

## Materials and methods

**Key resources table**

| Reagent type (species) or resource | Designation | Source or reference | Identifiers | Additional information |
|---|---|---|---|---|
| Genetic reagent (*Drosophila melanogaster*) | Armenia[60] | *Drosophila* Genomics Resource Center | DGRC #103394 | Line14 is an isofemale line derived from Armenia[60] |
| Genetic reagent (*D. melanogaster*) | *nAChRα6* T2A Gal4 | Bloomington *Drosophila* Stock Center | BDSC #76137 | RRID:BDSC_76137 |
| Genetic reagent (*D. melanogaster*) | UAS-GFP.nls | Bloomington *Drosophila* Stock Center | BDSC #4775 | RRID:BDSC_4775 |

*Continued on next page*

*Continued*

| Reagent type (species) or resource | Designation | Source or reference | Identifiers | Additional information |
| --- | --- | --- | --- | --- |
| Genetic reagent (*D. melanogaster*) | mito-roGFP2-Orp1 | Bloomington *Drosophila* Stock Center | BDSC #67672 | RRID:BDSC_67672 |
| Genetic reagent (*D. melanogaster*) | UAS-Sod2 | Bloomington *Drosophila* Stock Center | BDSC #24494 | RRID:BDSC_24494 |
| Genetic reagent (*D. melanogaster*) | UAS-Sod1 | Bloomington *Drosophila* Stock Center | BDSC #24750 | RRID:BDSC_24750 |
| Genetic reagent (*D. melanogaster*) | Elav-Gal4 | Bloomington *Drosophila* Stock Center | BDSC #458 | RRID:BDSC_458 |
| Genetic reagent (*D. melanogaster*) | Canton-S | Bloomington *Drosophila* Stock Center | BDSC #64349 | RRID:BDSC_64349 |
| Genetic reagent (*D. melanogaster*) | Canton-S *Dα6 KO*; Canton-S *nAChRα6 knockout* | This paper | | Mutant strain generated by CRISPR and maintained in T. Perry Lab |
| Genetic reagent (*D. melanogaster*) | Line14 *Dα6* loss-of-function mutant; Line14 *Dα6nx* | *Perry et al., 2015* (doi:10.1016/j.ibmb.2015.01.017) | | Mutant strain generated by EMS and maintained in T. Perry Lab |
| Genetic reagent (*D. melanogaster*) | GCaMP5G:tdTomato cytosolic [$Ca^{2+}$] sensor | Bloomington *Drosophila* Stock Center | BDSC #80079 | RRID:BDSC_80079 |
| Chemical compound, drug | Spinosad | Sigma-Aldrich | Product #33706 | |
| Chemical compound, drug | Antioxidant N-acetylcysteine amide; NACA | *Liu et al., 2015* (doi:10.1016/j.cell.2014.12.019) | | Provided by Hugo J. Bellen Lab |
| Chemical compound, drug | DHE | Sigma-Aldrich | Product #D7008 | |
| Chemical compound, drug | Nile Red | Sigma-Aldrich | Product #N3013 | |
| Chemical compound, drug | LysoTracker Red DND-99 (1:10,000) | Invitrogen | Cat #L7528 | |
| Commercial assay, kit | Mitochondrial aconitase activity kit | Sigma-Aldrich | Product #MAK051 | |
| Commercial assay, kit | ATP assay kit | Abcam | Product #ab83355 | |

## Fly strains and rearing

Line14 (*Perry et al., 2008*), an isofemale line derived from Armenia[60] (currently named Aashtrak, Drosophila Genomics Resource Center #103394), was used as the susceptible wild-type line for all assays except the following. GCaMP experiment: UAS-tdTomato-P2A-GCaMP5G (III) (*Daniels et al., 2014*; *Wong et al., 2014*) was crossed with *nAChRα6* T2A Gal4 (BDSC #76137). Expression of *nAChRα6* gene in adult brains: *nAChRα6* T2A Gal4 (BDSC #76137) was crossed with UAS-GFP.nls (BDSC #4775). Insecticide impact on mitochondrial turnover: the MitoTimer line (*Laker et al., 2014*) was used. Insecticide impact on mitochondrial ROS generation: the mito-roGFP2-Orp1 line (BDSC #67672) (*Albrecht et al., 2011*) was used. Overexpressing Sod2 and Sod1 in the central nervous system: UAS-Sod2 strain (BDSC #24494) and UAS-Sod1 strain (BDSC #24750) were crossed with a

Elav-Gal4 driver (BDSC #458). GCaMP experiment: UAS-tdTomato-P2A-GCaMP5G (III) (*Daniels et al., 2014*; *Wong et al., 2014*) was crossed with *nAChRα6* T2A Gal4 (BDSC #76137). Two mutants for the *nAChRα6* gene, which confers resistance to spinosad (*Perry et al., 2015*) and their background control lines, were used to dissect the differences between phenotypes caused by spinosad mode of action and phenotypes caused exclusively by *nAChRα6* loss of function. The first of these is Line14 *Dα6^{nx}* strain, a loss-of-function mutant recovered following EMS mutagenesis in the Line14 genetic background, with no detectable *nAChRα6* expression (*Perry et al., 2015*). The second mutant is a CRISPR knockout of *nAChRα6* generated in the Canton-S genetic background (*Luong, 2018*). For experiments aiming to investigate the trafficking of nAChRα6 in brains, UAS *Dα6* CFP tagged strain built in Line14 *Dα6^{nx}* background was crossed to a native Dα6 driver (Gal4-L driver) in the same background (*Perry et al., 2015*). For experiments involving larvae, flies were reared on standard food media sprinkled with dried yeast and maintained at 25°C. Early third-instar larvae were used in all experiments involving larval stage. For experiments involving adults, flies were reared in molasses food and maintained at 25°C. In all experiments involving adult flies, only virgin females were used to maintain consistency.

## Generation of CFP tagged *Dα6* subunit

To create the CFP tagged *Dα6* subunit for expression, a sequential PCR strategy was used to introduce the tag within the TM3-TM4 cytoplasmic loop region. Amplification of the pCyPet-His plasmid (Addgene #14030) with primers adding Gly-Ala-Gly and flanking homology arms to the *Dα6* insertion site (A385F_CFP_YFP and A385R_CFP) was performed. This fragment was purified and combined in a PCR reaction with a wild-type cDNA clone of *Dα6* (*Perry et al., 2015*) and reverse primer (da6_cloneR) to produce a fusion product. This fusion fragment was purified and combined in a PCR reaction with a wild-type cDNA clone of *Dα6* (*Perry et al., 2015*) and forward primer (da6_cloneF) to amplify a fragment encoding the full-length *Dα6* protein with an incorporated CFP tag (Dα6CFP – sequence provided below), and this was cloned into pUAST (*Bischof et al., 2007*). Following this, as per *Nguyen et al., 2021*, transgenic flies in the correct genetic background were made using microinjection into *y w M{eGFP.vas-int.Dm}ZH-2A; Dα6^{nx}; M{RFP.attP}ZH-86Fb* (*Bischof et al., 2007*; *Nguyen et al., 2021*), *cre*-recombinase was used to excise the 3XP3 RFP and miniwhite regions of the genomic insertion flanked by *lox-P* sites, and to control expression in this study, Dα6^{nx}; Dα6^{CFP} flies were crossed to a native Dα6 GAL4 driver *w; Dα6>GAL4; Dα6^{nx}* (*Perry et al., 2015*).

## Primer sequences

```
>A385F_CFP_YFP
CCTCCAAATCCCTGCTGGCCGGAGCAGGAATGTCTAAAGGTGAAGAATT
>A385R_CFP
TCGTCGATGTCGAGGACATTTCCTGCTCCTTTGTACAATTCATCCATAC
>da6_cloneF
GTAGCCATTCAACCCGAGAG
>da6_cloneR
GCTTCCGACGTATCCGTAGC
Dα6CFP – nucleotide sequence
GTAGCCATTCAACCCGAGAGCCACGCGATACAAACAAGCCAAGGACATGGACTCCCCGCT
GCCAGCGTCGCTGTCGCTGTTTGTCCTGTTGATCTTTCTGGCGATAATTAAAGAAAGCTG
TCAAGGACCTCATGAAAAGCGCCTGCTGAACCATCTGCTGTCCACCTACAATACGCTGGA
GCGACCCGTGGCCAATGAATCGGAGCCCCTGGAGATTAAGTTCGGACTGACGCTGCAGCA
GATCATCGACGTGGACGAGAAGAATCAGCTTCTCATAACGAATCTTTGGCTTTCGTTGGA
GTGGAACGACTACAATCTGCGCTGGAATGAAACGGAATACGGCGGGGTCAAGGATCTACG
AATCACGCCCAACAAGCTGTGGAAGCCCGACGTGCTCATGTACAACAGCGCGGATGAGGG
ATTCGATGGCACGTATCACACCAACATTGTGGTCAAACATGGCGGCAGTTGTCTGTACGT
GCCCCCTGGTATCTTCAAGAGCACATGCAAGATGGACATCACGTGGTTCCCATTTGATGA
CCAACATTGCGAAATGAAATTCGGTAGTTGGACTTACGATGGAAATCAGTTGGATTTGGT
TTTGAGTTCCGAAGATGGAGGGGATCTTTCCGATTTCATAACAAATGGCGAGTGGTACTT
GCTTGCCATGCCGGGAAAGAAGAATACGATAGTCTACGCCTGCTGCCCAGAACCATATGT
CGATATCACCTTTACTATACAAATTCGTCGCCGTACATTATATTATTTTTTCAATTTAATCGTG
CCATGTGTGCTAATCTCATCGATGGCCCTACTGGGCTTCACATTGCCGCCGGATTCGGGC
GAGAAACTGACGCTGGGAGTTACAATTCTTCTATCGCTCACAGTGTTTCTCAACCTTGTA
```

```
GCTGAGACATTGCCCCAAGTATCTGATGCAATCCCCTTGTTAGGCACCTACTTCAATTGC
ATCATGTTCATGGTCGCATCGTCGGTGGTGCTGACAGTAGTGGTGCTCAACTACCACCAT
CGCACAGCGGACATTCACGAGATGCCACCGTGGATCAAGTCCGTTTTCCTACAATGGCTG
CCCTGGATCTTGCGAATGGGTCGACCCGGTCGCAAGATTACACGCAAAACAATACTATTA
AGCAATCGCATGAAGGAGCTGGAGCTAAAGGAGCGCCCCTCCAAATCCCTGCTGGCCGGA
GCAGGAATGTCTAAAGGTGAAGAATTATTCGGCGGTATCGTCCCAATTTTAGTTGAATTA
GAGGGTGATGTTAATGGTCACAAATTTTCTGTCTCCGGTGAAGGTGAAGGTGATGCTACG
TACGGTAAATTGACCTTAAAATTTATTTGTACTACTGGTAAATTGCCAGTTCCATGGCCAACCT
TAGTCACTACTCTGACTTGGGGTGTTCAATGTTTTTCTAGATACCCAGATCATATGAAACAACA
TGACTTTTTCAAGTCTGTCATGCCAGAAGGTTATGTTCAAGAAGAACTATTTTTTTCAAAGAT
GACGGTAACTACAAGACCAGAGCTGAAGTCAAGTTTGAAGGTGATACCTTAGTTAATAGA
ATCGAATTAAAAGGTATTGATTTTAGAGAAGATGGTAACATTTTAGGTCACAAATTGGAATACA
ACTATATCTCTCACAATGTTTACATCACCGCTGACAAACAAAAGAATGGTATCAAAGCTA
ACTTCAAAGCCAGACACAACATTACCGATGGTTCTGTTCAATTAGCTGACCATTATCAAC
AAAATACTCCAATTGGTGATGGTCCAGTCATCTTGCCAGACAACCATTACTTATCCACTC
AATCTGCCTTATCTAAAGATCCAAACGAAAAGAGAGACCACATGGTCTTGCTCGAATTTG
TTACTGCTGCTGGTATTACCCATGGTATGGATGAATTGTACAAAGGAGCAGGAAATGTCC
TCGACATCGACGACGACTTTCGGCACACAATATCTGGCTCCCAAACCGCCATTGGCTCGT
CGGCCAGCTTCGGTCGGCCCACAACGGTGGAGGAGCATCACACGGCCATCGGCTGCAATC
ACAAAGATCTTCATCTAATTCTTAAAGAATTGCAATTTATTACGGCGCGGATGCGCAAAG
CTGACGACGAAGCGGAATTGATCGGCGATTGGAAGTTCGCGGCAATGGTTGTGGATAGAT
TTTGTTTAATTGTTTTCACGCTCTTCACGATTATTGCAACGGTTACGGTGCTGCTCTCCGCTCC
GCACATAATCGTGCAATAAGGACGCTCGAATTAGGCCATTAAGCTACGGATACGTCGGAAGC
```

## Insecticide dilution and exposure

Pure spinosad (Sigma-Aldrich) was used in all assays. The chemical was diluted to create 1000 ppm stocks solution, using dimethyl sulfoxide (DMSO), and was kept on freezer (–20°C). Before exposures, 5× stocks were generated for the dose being used by diluting the 1000 ppm stock in 5% Analytical Reagent Sucrose (Chem Supply) solution (or equivalent dose of DMSO for controls).

## Antioxidant treatment

The antioxidant, NACA, was used as previously described (*Martelli et al., 2020*). Briefly, larvae were treated with 300 μg/mL of NACA in 5% Analytical Reagent Sucrose (Chem Supply) solution for 5 hr prior to exposure to spinosad exposures.

## Fly media used

| Standard food (1 L) | | Apple juice plates (1 L) | | Molasses food (1 L) | |
|---|---|---|---|---|---|
| H$_2$O | 987 mL | H$_2$O | 720 mL | H$_2$O | 800 mL |
| Potassium tartrate | 8.0 g | Agar | 20 g | Molasses | 160 mL |
| Calcium chloride | 0.5 g | Apple juice | 200 mL | Maize meal | 60 g |
| Agar | 5.0 g | Brewer's yeast | 7.0 g | Dried active yeast | 15 g |
| Yeast | 12 g | Glucose | 52 g | Agar | 6.0 g |
| Glucose | 53 g | Sucrose | 26 g | Acid mix | 7.5 mL |
| Sucrose | 27 g | Tegosept | 6.0 mL | Tegosept | 5.0 mL |
| Semolina | 67 g | | | | |
| Acid mix | 12 mL | | | | |
| Tegosept | 15 mL | | | | |

## Larval movement assay

Larval movement in response to insecticide exposure was quantified by Wiggle Index Assay, as described by *Denecke et al., 2015*. 25 third-instar larvae were used for a single biological replicate and 4 replicates were tested for each exposure condition. Undosed larvae in NUNC cell plates

(Thermo Scientific) in 5% Analytical Reagent Sucrose (Chem Supply) solution were filmed for 30 s, and then 30 min, 1 hr, 1 hr, and 30 min and 2 hr after spinosad exposure. The motility at each time point is expressed in terms of relative movement ratio (RMR), normalized to motility prior to spinosad addition.

## Larval viability and adult survival tests

For all tests, five replicates of 20 individuals (100 individuals) per condition were used. In assessing third-instar larval viability and metamorphosis following insecticide exposure, individuals were rinsed three times with 5% w/v sucrose (Chem Supply) and placed in vials on insecticide-free food medium. Differences between adult eclosion rates were analyzed using Student's unpaired $t$-test. Correct percentage survival of larvae exposed to 0.5 ppm spinosad for 2 hr, or 0.1 ppm spinosad for 4 hr, was analyzed using Abbot's correction. To examine the survival of adult flies chronically exposed to 0.2 ppm spinosad, five replicates of 20 females (3–5 days old) were exposed for 25 days. The same number of flies was used for the control group. Statistical analysis was based on the Kaplan–Meier method, and data were compared by the log-rank Mantel–Cox test.

## GCaMP assay

Cytosolic [$Ca^{2+}$] in *Drosophila* primary neurons was measured as previously described (*Martelli et al., 2020*). Briefly, four brains from third-instar larvae were dissected to generate ideal number of cells for three plates. Cells were allowed to develop in culture plates (35 mm glass-bottom dishes with 10 mm bottom well [Cellvis], coated with concanavalin A [Sigma]) with Schneider's media for 4 days with the media refreshed daily. Recording was done using a Nikon A1 confocal microscope, ×40 air objective, sequential 488 nm and 561 nm excitation. Measurements were taken at 3 s intervals. Cytosolic $Ca^{2+}$ levels were reported as GCaMP5G signal intensity divided by tdTomato signal intensity. Signal was recorded for 60 s before the addition of 2.5 ppm or 25 ppm spinosad to the bath solution. 5 min after that, both insecticide and control groups were stimulated by the cholinergic agonist carbachol (100 µM) added to the bath solution, and finally, the SERCA inhibitor thapsigargin (5 µM) was added after a further 1 min. At least 50 neuronal cells were evaluated per treatment. The data were analyzed using one-way ANOVA followed by Tukey's honestly significant difference test.

## Evaluation of mitochondrial turnover

Mitochondrial turnover was assessed as previously described (*Martelli et al., 2020*). Larvae of the MitoTimer line were exposed to 2.5 ppm spinosad for 2 hr. Control larvae were exposed to 2.5 ppm DMSO. Midguts and brains were dissected in PBS and fixed in 4% paraformaldehyde (PFA) (Electron Microscopy Sciences) and mounted in VECTASHIELD (Vector Laboratories). 20 anterior midguts and 20 pairs of optical lobes were analyzed for each condition. Confocal microscopy images were obtained with a Leica SP5 Laser Scanning Confocal Microscope at ×200 magnification for both green (excitation/emission 488/518 nm) and red (excitation/emission 543/572 nm) signals. Three independent measurements along the z stack were analyzed for each sample. Fluorescence intensity was quantified on ImageJ software, and data were analyzed using one-way ANOVA followed by Tukey's honestly significant difference test.

## Evaluation of mitochondrial ROS generation using Mito-roGFP2-Orp1

The mito-roGFP2-Orp1 (BDSC #67672) was used to measure the production of mitochondrial $H_2O_2$ (Albrecht et al., 2011). Larvae were exposed to 2.5 ppm spinosad for 2 hr (controls exposed to 2.5 ppm DMSO). Anterior midguts and brains were dissected in Schneider's media (Gibco) and immediately mounted in VECTASHIELD (Vector Laboratories) for image acquisition. Confocal microscopy images were obtained with a Leica SP5 Laser Scanning Confocal Microscope under excitation/emission 488/510 nm (reduced) or 405/510 nm (oxidized). Three independent measurements along the z stack were analyzed for each sample. Fluorescence intensity was quantified on ImageJ software, and data were analyzed using Student's unpaired $t$-test.

## Systemic mitochondrial aconitase activity

Relative mitochondrial aconitase activity was quantified using the colorimetric Aconitase Activity Assay Kit from Sigma (#MAK051), following the manufacturer's instructions as previously described (*Martelli*

*et al., 2020*). A total of six biological replicates (25 whole larvae per replicate) were exposed to 2.5 ppm spinosad for 2 hr, whilst six control replicates (25 whole larvae per replicate) were exposed to DMSO for 2 hr. Absorbance was measured at 450 nm in a FLUOstar OPTIMA (BMG LABTECH) microplate reader using the software OPTIMA and normalized to sample weight. The data were analyzed using one-way ANOVA followed by Tukey's honestly significant difference test.

### Systemic ATP levels

Relative ATP levels were quantified fluorometrically using an ATP assay kit (Abcam #83355), following the manufacturer's instructions as previously described (*Martelli et al., 2020*). A total of six biological replicates (20 larvae per replicate) were exposed to 2.5 ppm spinosad for 2 hr, whilst six control replicates (20 larvae per replicate) were exposed to DMSO for 2 hr. Fluorescence was measured as excitation/emission = 535/587 nm in FLUOstar OPTIMA (BMG LABTECH) microplate reader using the software OPTIMA and normalized to sample weight. The data were analyzed using one-way ANOVA followed by Tukey's honestly significant difference test.

### Measurement of superoxide ($O_2^-$) and other ROS levels

Levels of superoxide and other ROS were assessed by DHE staining (Sigma-Aldrich), as described in *Owusu-Ansah et al., 2008*. Briefly, larvae were dissected in Schneider's media (Gibco) and incubated with DHE at room temperature on an orbital shaker for 7 min in dark. Tissues were fixed in 8% PFA (Electron Microscopy Sciences) for 5 min at room temperature on an orbital shaker in dark. Tissues were then rinsed with PBS (Ambion) and mounted in VECTASHIELD (Vector Laboratories). Confocal microscopy images were obtained with a Leica SP5 Laser Scanning Confocal Microscope at ×200 magnification (excitation/emission 518/605 nm). Fluorescence intensity was quantified on ImageJ software, and data were analyzed using one-way ANOVA followed by Tukey's honestly significant difference test.

### Evaluation of lipid environment of metabolic tissues in larvae

Fat bodies and Malpighian tubules were dissected in PBS (Ambion) and subjected to lipid staining with Nile Red N3013 Technical grade (Sigma-Aldrich) as previously described (*Martelli et al., 2020*). Three biological replicates were performed for each exposure condition, each replicate consisting of a single tissue from a single larva. Tissues were fixed in 4% PFA (Electron Microscopy Sciences) and stained with 0.5 µg/mL Nile Red/PBS for 20 min in dark. Slides were mounted in VECTASHIELD (Vector Laboratories) and analyzed using a Leica SP5 Laser Scanning Confocal Microscope at ×400 magnification. Red emission was observed with 540 ± 12.5 nm excitation and 590 LP nm emission filters. Images were analyzed using ImageJ software. For fat bodies, the number, size, and percentage of area occupied by LDs were measured in five different random sections of 2500 µm$^2$ per sample (three samples per group). For Malpighian tubules, the number of LDs was measured in five different random sections of 900 µm$^2$ per sample (three samples per group). The data were analyzed using one-way ANOVA followed by Tukey's honestly significant difference test.

### Lipid quantification in larval hemolymph

Extracted hemolymph lipids were measured using the sulfo-phospho-vanillin method (*Cheng et al., 2011*) as previously described (*Martelli et al., 2020*). 30 third-instar larvae were used for a single biological replicate, and 7 replicate samples were prepared for each exposure condition. Absorbance was measured at 540 nm in a CLARIOstar (BMG LABTECH) microplate reader using MARS Data Analysis Software (version 3.10R3). Cholesterol (Sigma-Aldrich) was used for the preparation of standard curves. The data were analyzed using one-way ANOVA followed by Tukey's honestly significant difference test.

### Lipid extraction and analysis using liquid chromatography-mass spectrometry

Lipidomic analyses of whole larvae exposed for 2 hr to 2.5 ppm spinosad were performed in biological triplicate and analyzed by electrospray ionization-mass spectrometry (ESI-MS) using an Agilent Triple Quad 6410 as previously described (*Martelli et al., 2020*). Briefly, samples were transferred to Cryo-Mill tubes treated with 0.001% butylated hydroxytoluene (BHT) and frozen in liquid nitrogen. Samples

were subsequently homogenized using a CryoMill (Bertin Technologies) at −10°C. Then, 400 µL of chloroform was added to each tube and samples were incubated for 15 min at room temperature in a shaker at 1200 rpm. Samples were then centrifuged for 15 min at 13,000 rpm at room temperature; the supernatants were removed and transferred to new 1.5 mL microtubes. For a second wash, 100 µL of methanol (0.001% BHT and 0.01 g/mL 13C5 valine) and 200 µL of chloroform were added to CryoMill tubes, followed by vortexing and centrifugation as before. Supernatants were transferred to the previous 1.5 mL microtubes. A total of 300 µL of 0.1 M HCl was added to pooled supernatants, and microtubes were then vortexed and centrifuged (15 min, room temperature, 13,000 rpm). Upper phases (lipid phases) were collected and transferred to clean 1.5 mL microtubes, as well as the lower phases (polar phases). All samples were kept at −20°C until analysis. For liquid chromatography-mass spectrometry (LC-MS) analysis, microtubes were shaken for 30 min at 30°C, then centrifuged at 100 rpm for 10 min at room temperature after which the supernatants were transferred to LC vials. Extracts were used for lipid analysis. For statistical analysis, the concentration of lipid compounds was initially normalized to sample weight. Principal components analysis (PCA) was calculated to verify the contribution of each lipid compound in the variance of each treatment. PCA was calculated using the first two principal component axes. To discriminate the impacts of spinosad on the accumulation of specific lipid compounds, we performed a one-way ANOVA followed by Tukey's honestly significant difference test.

## Investigating impacts on lysosomes

To investigate spinosad impacts on lysosomes, the LysoTracker staining was used on larval brains. Larvae were dissected in PBS and tissue immediately transferred to PBS solution containing LysoTracker Red DND-99 (1:10,000) (Invitrogen) for 7 min. Tissues were then rinsed three times in PBS, and slides were mounted for immediate microscopy at ×400 magnification (DsRed filter) with a Leica SP5 Laser Scanning Confocal Microscope. To investigate the hypothesis of Dα6 nAChRs being endocytosed and digested by lysosomes after exposure to spinosad, brains from larvae obtained by crossing UAS *Dα6* CFP tagged (in Line14 *Dα6^{nx}* background) to Gal4-L driver (in Line14 *Dα6^{nx}* background) were also subjected to LysoTracker staining. Images were analyzed using the software ImageJ, and data were analyzed using one-way ANOVA followed by Tukey's honestly significant difference test.

## Electrophysiology of the retina

Amplitudes and on-transients were assessed as previously described (*Martelli et al., 2020*). Briefly, adult flies were anesthetized and glued to a glass slide. A reference electrode was inserted in the back of the fly head and the recording electrode was placed on the corneal surface of the eye, both electrodes were filled with 100 mM NaCl. Flies were maintained in the darkness for at least 5 min prior to a series of 1 s flashes of white light delivered using a halogen lamp. During screening, 8–10 flies per treatment group were tested. For a given fly, amplitude and on-transient measurements were averaged based on the response to the three light flashes. Responses were recorded and analyzed using AxoScope 8.1. The data were analyzed using one-way ANOVA followed by Tukey's honestly significant difference test.

## Nile Red staining of adult retinas

For whole-mount staining of fly adult retinas, heads were dissected in cold PBS (Ambion) and fixed in 37% formaldehyde overnight. Subsequently, the retinas were dissected and rinsed several times with 1× PBS and incubated for 15 min at 1:1,000 dilution of PBS with 1 mg/mL Nile Red (Sigma). Tissues were then rinsed with PBS and immediately mounted with VECTASHIELD (Vector Labs) for same-day imaging. For checking the effects of chronic exposures, eight retinas from eight adult female flies were analyzed per treatment/genotype per time point. Images were obtained with a Leica TCS SP8 (DM600 CS), software LAS X, ×600 magnification, and analyzed using ImageJ. The data were analyzed using one-way ANOVA followed by Tukey's honestly significant difference test.

## Expression of Dα6 nAChRs in the brain

The expression pattern of nAChR-Dα6 gene in adult brains was assessed in the crossing between Dα6 T2A Gal4 (BDSC #76137) and UAS-GFP.nls (BDSC #4775). Adult brains were fixed in 4% PFA (Electron Microscopy Sciences) in PBS for 20 min at room temperature. PFA was removed and tissues were

washed three times in PBS. Samples were mounted in VECTASHIELD (Vector Laboratories). Images were obtained with a Leica TCS SP8 (DM600 CS), software LAS X, ×400 magnification, using GFP channel. Images were analyzed using the software ImageJ.

### Adult brain histology (H&E staining)

Adult fly heads were fixed in 8% glutaraldehyde (EM grade) and embedded in paraffin. Sections (10 µm) were prepared with a microtome (Leica) and stained with H&E as described (*Chouhan et al., 2016*). At least three animals were examined for each group (20 days exposure to 0.2 ppm spinosad plus control group) in terms of percentage of brain area vacuolated. The data were analyzed using Student's unpaired *t*-test.

### Transmission electron microscopy (TEM)

Laminas of adult flies chronically exposed to 0.2 ppm spinosad for 20 days (controls exposed to equivalent volume of DMSO) were processed for TEM imaging as described (*Luo et al., 2017*). TEM of laminas of 20-day-old Canton-S and Canton-S *Dα6 KO* mutants aged in the absence of spinosad was also investigated. Samples were processed using a Ted Pella Bio Wave microwave oven with vacuum attachment. Adult fly heads were dissected at 25°C in 4% PFA, 2% glutaraldehyde, and 0.1 M sodium cacodylate (pH 7.2). Samples were subsequently fixed at 4°C for 48 hr. 1% osmium tetroxide was used for secondary fixation with subsequent dehydration in ethanol and propylene oxide. Samples were then embedded in Embed-812 resin (Electron Microscopy Sciences, Hatfield, PA). 50-nm ultra-thin sections were obtained with a *Leica UC7* microtome and collected on Formvar-coated copper grids (Electron Microscopy Sciences). Specimens were stained with 1% uranyl acetate and 2.5% lead citrate and imaged using a JEOL JEM 1010 transmission electron microscope with an AMT XR-16 mid-mount 16 megapixel CCD camera. For quantification of ultrastructural features, electron micrographs were examined from three different animals per treatment. The data were analyzed using Student's unpaired *t*-test.

### Bang sensitivity

The BS phenotype was tested after 1, 10, and 20 days of chronic exposure to 0.2 ppm spinosad. Flies were vortexed on a VWR vortex at maximum strength for 10 s. The time required for flies to flip over and regain normal standing posture was then recorded. The data were analyzed using Kruskal–Wallis followed by Dunn's multiple comparisons test.

### Climbing assay

Climbing phenotype was tested after 1, 10, and 20 days of exposure to 0.2 ppm spinosad. Five adult female flies were placed into a clean vial and allowed to rest for 30 min. Vials were tapped against a pad, and the time required for the flies to climb up to a predetermined height (7 cm) was recorded. Flies that did not climb the predetermined height within 30 s were deemed to have failed the test. The data were analyzed using Kruskal–Wallis followed by Dunn's multiple comparisons test.

### Graphs and statistical analysis

Wiggle Index analyses were performed using the software R (v.3.4.3) (*Denecke et al., 2015*). All other graphs and statistical analyses were performed using GraphPad Prism (v.9.2.0). Image panels were designed using the free image software Inkscape (0.92.4).

Many of the analyses performed here were conducted on spinosad and imidacloprid in parallel with these treatments sharing the same controls, allowing direct comparison of the impact of these insecticides. The imidacloprid data are published in *Martelli et al., 2020*. The data with shared wild-type control flies (unexposed) are shown in *Figures 1A, D, E and 3A–D*, *Figures 4A–C and 5A, B, C, D, G*, *Figure 5—figure supplements 1 and 2*, *Figure 6*, *Figure 6—source data 1*, *Figures 7A–C and 9B and C*.

## Acknowledgements

The pCyPet-His plasmid was a gift from Patrick Daugherty (Addgene plasmid # 14030). FM was supported by a Victorian Latin America Doctoral Scholarship, an Alfred Nicholas Fellowship, a UoM Faculty of Science Travelling Scholarship, and The Robert Johanson and Anne Swann Fund – Native

Animals Trust (awarded to FM and TP). PB was supported by the University of Melbourne. HJB was supported by the Howard Hughes Medical Institute (HHMI) and is an investigator of HHMI. KV was supported by NIH (NIA) grant. Lipid analyses were performed at Metabolomics Australia at University of Melbourne, which is a National Collaborative Research Infrastructure Strategy initiative under Bioplatforms Australia Pty Ltd (http://www.bioplatforms.com/).

## Additional information

### Competing interests

Hugo J Bellen: Reviewing editor, *eLife*. The other authors declare that no competing interests exist.

### Funding

| Funder | Grant reference number | Author |
| --- | --- | --- |
| Howard Hughes Medical Institute | | Hugo J Bellen |
| Victoria State Government | Latin America Doctoral Scholarship | Felipe Martelli |
| University of Melbourne | Alfred Nicholas Fellowship | Felipe Martelli |
| University of Melbourne | Faculty of Science Travelling Scholarship | Felipe Martelli |
| The Robert Johanson and Anne Swann Fund | Native Animals Trust | Felipe Martelli Trent Perry |
| University of Melbourne | | Philip Batterham |
| National Institute on Aging | | Kartik Venkatachalam |

The funders had no role in study design, data collection and interpretation, or the decision to submit the work for publication.

### Author contributions

Felipe Martelli, Conceptualization, Data curation, Formal analysis, Investigation, Validation, Visualization, Writing – original draft, Writing – review and editing; Natalia H Hernandes, Ching-On Wong, Formal analysis, Investigation, Writing – review and editing; Zhongyuan Zuo, Formal analysis, Investigation, Visualization, Writing – review and editing; Julia Wang, Ute Roessner, Charles Robin, Investigation, Writing – review and editing; Nicholas E Karagas, Formal analysis, Investigation; Thusita Rupasinghe, Data curation, Formal analysis, Investigation; Kartik Venkatachalam, Funding acquisition, Investigation, Writing – review and editing; Trent Perry, Conceptualization, Funding acquisition, Writing – review and editing; Philip Batterham, Hugo J Bellen, Conceptualization, Funding acquisition, Investigation, Supervision, Writing – review and editing

### Author ORCIDs

Felipe Martelli http://orcid.org/0000-0003-4783-9025
Natalia H Hernandes http://orcid.org/0000-0001-5644-6974
Kartik Venkatachalam http://orcid.org/0000-0002-3055-9265
Trent Perry http://orcid.org/0000-0002-8045-0487
Philip Batterham http://orcid.org/0000-0001-9840-9119
Hugo J Bellen http://orcid.org/0000-0001-5992-5989

### Decision letter and Author response

Decision letter https://doi.org/10.7554/eLife.73812.sa1
Author response https://doi.org/10.7554/eLife.73812.sa2

## Additional files

### Supplementary files
• Transparent reporting form

### Data availability
All data generated or analysed during this study are included in the manuscript and supporting file; Source Data file has been provided for Figure 6.

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
