## [Editor Report]

Insecticides have been implicated in the decline of beneficial insect species. The organic insecticide Spinosad has emerged as a alternative to synthetic insecticides and is thought to be less harmful to beneficial insects than synthetic insecticides. This article used the insect model Drosophila to analyze the impact of Spinosad. It reveals that low doses of Spinosad antagonize its neuronal target, the nicotinic acetylcholine receptor subunit alpha 6, affecting the physiology of Drosophila. This study reveals that although being organic and assumed to be less harmful, spinosad can have profound impact on non-target insects.

---

## [Decision Letter]

**Decision letter after peer review:**

[Editors’ note: the authors submitted for reconsideration following the decision after peer review. What follows is the decision letter after the first round of review.]

Thank you for submitting your work entitled "The organic insecticide spinosad trigger lysosomal defect, ROS driven lipid dysregulation and neurodegeneration in flies" for consideration by *eLife*. Your article has been reviewed by 3 peer reviewers, and the evaluation has been overseen by a Reviewing Editor and a Senior Editor. The reviewers have opted to remain anonymous.

We are sorry to say that, after consultation with the reviewers, we have decided that your work will not be considered further for publication by *eLife*. But they all agree that they are happy to reconsider a revised version that provides more mechanistic insights on the action of the insecticide.

In this manuscript, it is shown that low doses of spinosad are toxic to *Drosophila*, suggesting that traces of this organic pesticide in the environment can be damaging to other useful insect species, as with the neonicotinoid imidacloprid. It is not easy to assess the novelty and significance of this claim. Spinosad has been known for a long time to have a large spectrum of action, including many species of Lepidoptera and Diptera, like e.g. the house fly (Liu and Yue J Econ Entomol 2000). In addition, field studies will be asked in any cases to confirm the danger of this pesticide at low doses out of the laboratory.

Mechanistically, the authors propose that spinosad accumulates in lysosomes after binding to nAChRα6 and that this would progressively disrupt their function. Although this hypothesis is suggested by the disappearance of the α6 subunit, reported in another paper this year (Nguyen et al., Pest Manag Sci 2021) this, and lysosomal swelling, is not directly demonstrated in this study. The authors also suggest that lysosome disruption would give rise to mitochondrial impairments, increased ROS production and finally neurodegeneration, but this causal link is hypothetical at this stage, as noted by some reviewers. The authors finally suggest that a toxic signal released from the brain propagates to peripheral tissues in spinosad–intoxicated flies, explaining the non–nervous system defect, but again, this idea is not addressed experimentally. While the topic was found to be of general interest, the reviewers also feel that this manuscript is very descriptive and it does not report enough original breakthrough findings to warrant publication in *eLife* at this stage. A new submission that includes more mechanistic insights might be considered for *eLife* but requires substantial improvement.

*Reviewer #2:*

This article reports that an organic insecticide used at low doses creates nevertheless considerable havoc, including extensive neurodegeneration in adults. The authors show that exposure to spinosad blocks acetylcholine–triggered calcium influx in larval neurons that express the Acetylcholine receptor (AchR) subunit Dalpha6, the target of spinosad. They document that exposure to spinosad leads to a Dalpha6–dependent enlargement of lysosomes in the nervous system where the Dalpha6 subunits accumulates, an increase in reactive oxygen species (ROS) generation in the brain, and a redistribution of lipids in the fat body from large lipid droplets (LDs) to numerous smaller ones, an effect that can be counteracted by pre–exposure to an antioxidant compound. The chronic exposure of adults to low spinosad doses leads to progressively altered behaviors as well as extensive neurodegeneration in the visual system as well as in parts of the central brain.

The strengths of the study lie in the description of several damages induced by exposure to spinosad. The part on the neurodegeneration in adults is especially impressive. The central message of the authors that even organic pesticides have unintended severe effects to a nontarget insect upon exposure to low doses (chronic low doses in the case of adults) is thus supported by experimental evidence, which however might be improved in specific cases.

The major weaknesses of the study are that the authors hardly investigate causal relationships between the different phenotypes they observe and remain too descriptive. For instance, their preferred model is that lysosomal disorders are at the origin of ROS production in mitochondria, with no experimental evidence to back up this claim. This reviewer is also not thoroughly convinced that ROS are produced by mitochondria, as the decreased activity of aconitase and impact on ATP production might be mediated by another mechanism, e.g., increased mitophagy. They do not consider the alternative possibility that lysosomal dysfunction might be caused by ROS production. Thus, one would like to know whether the expansion of lysosomes still occurs when ROS production is inhibited. This would be best achieved using two independent approaches, one based on exposure to reducing agents (do they have any effect on their own (control missing)?), and the other based on genetic strategies that however require identifying the enzyme(s) responsible for ROS production.

They also do not document whether the ROS phenotype depends on the presence of the Dalpha6 subunit of the AchR. It would also be positive to check whether there is any altered production of ROS occurring directly in the fat body or Malpighian tubules. Can the authors exclude that the Dalpha6 nAchR subunit might be expressed at low levels in these tissues, for instance in the enteroendocrine cells of the gut?

One major question raised by this study is whether insects other than those targeted by this pesticide are exposed to low doses, for instance in fields neighboring the treated area and undergo damages similar to those reported here. Some of the techniques used in this study may be employed on other insects collected near fields treated with spinosad, such as examining the distribution of LDs in their fat bodies. In this respect, it might be useful determining whether this phenotype as well as the expanded lysosomes observed in larvae are also found in adults.

As stated in above, a major weakness of the study is a lack of functional connections made between the different phenotypes documented by the authors. While it may not be easy to determine whether ROS production depends on lysosome dysfunction, the reverse can be investigated relatively easily, for instance by using larvae pre–treated with NACA and other reagents as detailed below.

As regards ROS, one definitely would like to know whether they are generated solely in neurons expressing the Dalpha6 nAchR subunit or whether they are found in other neuronal cell types or glial cells. A subcellular resolution would be ideal. To this end, the authors might want to use ratiometric ROS stress reporters such as Mito–roGFP (Albrechts et al., Cell Metabolism, 2011). It would also be desirable to rely on more than one method to block ROS production. In terms of biochemical reducing agents, the authors might want to also check MitoTempo, which is more specifically acting on mitochondrial ROS. On the genetic side, the authors should overexpress Superoxide Dismutase 2 (SOD2), and possibly SOD1 as a control. Another useful tool is a mitochondrially–addressed catalase (UAS–mtcatalase) developed originally by William Orr, e.g., Mockett et al., Free Radic. Biol. Med. , 2003.

Once the mitochondrial origin of ROS is demonstrated, it will then be highly interesting to determine whether lysosome expansion still occurs when the source is inhibited.

The authors rely solely on Nile Red for assessing the distribution of LDs. A control with lipidTox staining, which is more specific for neutral lipids, would be appropriate. Also, while small LDs may increase the surface available for storing oxidized lipids, it would be useful to determine whether this is indeed the case by staining oxidized lipids. It is also not clear whether the overall levels of triglycerides are changed upon spinosad exposure and measuring the levels of TAGs biochemically would provide an interesting counterpoint to lipid stains.

What is the biological function of this redistribution of lipids in the fat body? Would the genetic inhibition of Dalpha6–expressing neurons lead to the LD phenotype (likely unlikely given the loss–of–function phenotype). Conversely, would genetically activating neuronal signaling in these neurons be able to reverse the effects of spinosad exposure, at least as regards the LD phenotype in the fat body?

*Reviewer #3:*

In this paper, the authors used *Drosophila* as a model to study the effect of the insecticide spinosad on the non–target insect. They investigated the histological, physiological, and behavioral impacts of spinosad and found that low dose administration of this pesticide causes neuronal lysosomal alterations and mitochondrial impairment and ROS elevation and lead to altered lipids profile and neurodegeneration and subsequent consequences. One of the main results of the study is to show that spinosad induce major systemic metabolic changes through its action in the brain. Since spinosad have been shown to greatly affect the non–pest insect population, the findings here are interesting and appealing to many researchers. I only have two major comments as listed below.

1. The authors suggest that spinosad actions on neurons expressing Dα6 nAChRs inducing subsequent lipid phenotypes, especially through ROS. They also observed an increased ROS level in tissues as gut. Although Dα6 mutant showed a decreased lipid droplets in fatbody, there is no direct evidence linking neuronal ROS with this phenotype. Thus, it remains unclear whether peripheral tissues ROS also contribute to the altered lipid phenotypes in the fly. Furthermore, whether elevated gut ROS level is induced by neuronal ROS burst or independent from neuronal phenotype is not clear. I would suggest using fly genetic tools (Gal4–uas system) to manipulate ROS level in the neurons and peripheral tissues to further elucidate the conclusion.

2. The author's finding pointed to the mechanisms of spinosad on non–target insects. However, since insects are the most diverse group of animals and there is a huge divergence between *Drosophila* and other insects, whether spinosad causes the same effects in other species needs to be clearly discussed.

3. The authors suggest that the spinosad effect on *Drosophila* was a consequence of its action on Dα6 nAChRs. It would be better to include survival data of Dα6 mutant upon Spinosad exposure to reinforce the conclusion.

4. It would also be interesting to see if there is also a ROS burst in the fatbody that leads to the metabolic phenotype directly. Overexpression of antioxidant enzymes (sod1, sod2, catalase) using elav–Gal4 (Pan neuron driver), myo–Gal4 (gut enterocytes driver) and c564–Gal4 (fatbody driver) could elucidate the causal relations between the neuronal phenotypes and lipid metabolic phenotypes.

*Reviewer #4:*

The manuscript by Felipe Martelli and coll. addresses the cellular and metabolic defects induced by poisoning with spinosad in the fruit fly *Drosophila*. Spinosad is a potent insecticide first identified as being produced by a soil bacterium and so considered as an organic product, which has been widely used in recent years. Published work showed that the main selective target of spinosad in insects is the α6 subunit of the nicotinic acetylcholine receptor (nAchR), which is selectively expressed in the nervous system in flies, and that knockout of nAChRα6 (Dα6) in *Drosophila* confers resistance to spinosad. This paper complements and extends these previous studies by showing that spinosad exposure triggers lysosomal swelling, mitochondrial impairment and oxidative stress in fly larval brain cells, and also induces a variety of metabolic dysregulation in peripheral tissues, including alterations in the larval lipidomic profile and an accumulation of small lipid droplets in the fat body. Pretreatment with the anti–oxidant N–acetylcysteine amide (NACA) alleviated some of these defects and partially, the larval lethality induced by spinosad. It is also shown that low doses of spinosad promoted neurodegeneration and loss of vision in adult female *Drosophila*.

Overall, this manuscript represents a substantial amount of experimental work that provides new insight into spinosad–induced toxicity. The authors introduce here several hypotheses to make sense of their various observations. Specifically, they propose that spinosad accumulates in lysosomes after binding to nAChRα6 and because of that would progressively disrupt lysosomal function. This could then give rise to mitochondrial dysfunction, increased ROS production and finally neurodegeneration. The authors also suggest that a toxic signal released from the brain after spinosad intoxication propagates to peripheral tissues, including the fat body, midgut, and Malpighian tubules, to promote disturbance in metabolism and lipid homeostasis. Although these hypotheses are not directly addressed experimentally in this study, they would certainly be interesting to explore in future work.

There are three sets of results for which I have concerns about some of the authors' conclusions:

– In Figure 1, it is not clear whether the lower carbachol–induced response in spinosad–treated cells reflects a decrease in calcium influx through nAChRs, or, alternatively, a reduced calcium release from intracellular stores (see e.g. Campusano et al. Dev Neurobiol 2007). This point has to be carefully considered as the lower response to carbachol could be either a direct or indirect effect of spinosad–induced Dα6 subunit internalization. I think that the reported experiments are not sufficient to decide between these possibilities.

– The second issue is the experiment of Figure 5, in which the authors showed that spinosad treatment of Dα6 KO flies had no effect on lipid level in the hemolymph, in contrast to the increase it induced in wild–type controls. This conclusion is apparently correct, but it does not take in account the fact that the lipid level actually appears at least as high in the Dα6 mutants as in their respective controls treated with spinosad. It seems therefore that the increase in lipid in the hemolymph is more likely caused by Dα6 deficiency than by spinosad treatment. Similarly, the percentage of the area occupied by LD in fat body of both Dα6 mutants is between that of untreated control flies and control flies treated with spinosad. This seems to be consistent with the idea that spinosad only exacerbates an increase in lipids which is for a large part induced by Dα6 deficiency. Incidentally, because Dα6 KO mutants are viable, this observation indicates that altered lipid homeostasis may not be a prominent cause of larval lethality in spinosad–exposed flies.

– My third main comment is that it would be very informative to compare the effects of spinosad in adult wild–type and Dα6 mutant flies, with respect to the survival, bang sensitivity and climbing ability tests (Figure 7), as well as for lipid deposits in the retina (Figure 8). These data would be rather easy to collect and would usefully complete these figures by indicating whether or not spinosad can induce defects in adult flies in the absence of Dα6. A positive answer would argue for the existence of other potential molecular targets of this insecticide.

I have provided more detail on these and other points below.

Main issues

1) Line 117 and 132: from experiments shown in Figure 1D, E, the authors conclude that spinosad exposure "prevent ca^2+^ flux into neurons expressing nAChRs" and so "blocks the function of Dα6–containing nAChRs". However, Campusano et al. Dev Neurobiol 2007, have shown that thapsigargin treatment reduces nicotine‐evoked calcium increase in Kenyon cells, consistent with a signal amplification by calcium release from intracellular stores. In addition, part of this calcium increase had a voltage‐gated calcium channel–dependent (VGCC) component, indicating that only a fraction of the calcium increase involves calcium influx directly through nAChRs. Therefore, the experiments in Figure 1D, E are not sufficient to decide, in my opinion, whether the reduction in carbachol–induced response in spinosad–treated cells reflects a decrease in calcium influx through nAChRs, or, alternatively, a reduced calcium release from intracellular stores. As mentioned in the public review, this point has to be carefully considered as the decreased response to carbachol could be either a direct or indirect effect of spinosad–induced Dα6 subunit internalization.

2) Line 263: "We also quantified the level of lipids in hemolymph. Whereas Line 14 and Canton S showed an average 10% and 13% increase in response to spinosad, respectively, neither of the Dα6 KO mutants showed significant changes (Figure 5E)." However, the lipid level appears to be at least as high in the Dα6 mutants as in their respective controls treated by spinosad (Figure 5E). It seems therefore that the increase in lipid in the hemolymph is more likely caused by Dα6 deficiency than by spinosad treatment. Similarly, the percentage of area occupied by LD in fat body of both Dα6 mutants is between that of untreated control flies and control flies treated with spinosad (compare Figure 4B and Figure 5B for the Line 14 background, and compare Figure 5C and Figure 5D for the Canton S background). This seems to be consistent with the idea that spinosad only exacerbates an increase in lipids which is largely induced by Dα6 deficiency. If this is correct, please amend the conclusions accordingly. For example, the sentence on page 25, line 479: "Dα6 knockout mutants exposed to spinosad show no accumulation of LD in the fat body or change of lipid levels in hemolymph indicating that these phenotypes are due to the spinosad:Dα6 interaction (Figure 5)" is not fully correct as it does not mention the fact that Dα6 deficiency by itself appears to increase LD accumulation.

3) It would be very informative to repeat the survival and behavior test experiments of Figure 7 in Dα6 mutant background. I was actually surprised that this was not included in the manuscript, as it would not involve a lot of work, and, importantly, it would indicate whether spinosad can induce behavior defects in the absence of Dα6, which would argue for the existence of other molecular targets for this toxin. I would recommend that lipid deposits in the retina (Figure 8A, B) be also quantified in Dα6 background for purpose of comparison.

– Line 125: "primary culture of neurons " could be "primary culture of third–instar larva brain neurons" to avoid misunderstanding.

– In Figure 1A, the statistical test used should be two–way ANOVA and not t–test, I think, as both the genotypes and time differ in each point, and in Figure 1E one–way ANOVA should be used, as the means of more than two groups were compared. Same issue in Figure 2B and in several other figures of the paper where the Student's t–test was systematically used.

– Figure 1B: I understand that spinosad precludes pupation of intoxicated larvae. However, after eight days, a wild–type third–instar larva should have become an adult fly. So, I guess that authors looked at fly survival, and not larval survival, for the wild–type. Please mention it, if it is right. In Figure 1C, did you treat the pupae or the larvae before pupal case formation with the toxin? This could be stated in the legend.

– Line 579: "derived from Armenia60 (*Drosophila* Genomics Resource Center #103394)" – I could not find this Armenia60 line referenced neither in the DGRC or Flybase sites. Please provide the correct source or link for this strain, and explain why it was used as a wild–type control.

– Line 581: "Expression of nAChR–Dα6 gene" – According to Flybase, this gene should be named: nAChRα6. The correct nomenclature should be mentioned at least once in the manuscript.

– Line 582: "the MitoTimer line (Gottlieb and Stotland, 2015) was used". The exact reference for this line is: Laker et al. J Biol Chem. (2014) 289(17):12005–15. Same correction required on line 187.

– Line 594: "In all experiments involving adult flies only females were used to maintain consistency." Adult *Drosophila* females have very different physiology compared to males. This is particularly relevant for reproductive organs, of course, but also for the gut and nervous system. Therefore, this particularity of this study is not negligible and should be mentioned in the abstract.

[Editors’ note: further revisions were suggested prior to acceptance, as described below.]

Thank you for resubmitting your work entitled "Low doses of the organic insecticide spinosad triggers lysosomal defects, elevated ROS, lipid dysregulation, and neurodegeneration in flies" for further consideration by *eLife*. Your revised article has been evaluated by Utpal Banerjee (Senior Editor) and Bruno LeMaitre (Reviewing Editor).

The manuscript by Martelli et al. has been clearly improved in this re-submission. In particular, the statistical analysis has been done more accurately and new interesting experiments have been added. But there are some remaining issues that need to be addressed, as outlined below:

Essential revisions:

1) While this revised version is definitely improved, a sticky point remains, namely that of the relationship between the lysosomal disorder and the production of ROS by mitochondria, which is not rigorously established. The authors use NACA to block the production of ROS. Since presumably NACA has a ubiquitous effect, the authors cannot formally exclude that the ROS originate from a source distinct from mitochondria. In this respect, it is noteworthy that the authors did not investigate the impact of NACA treatment on mitochondria. This reviewer does not understand the argument put forth by the authors as regards the lack of subcellular resolution of mito-roGFP reporters. Because the reporters are targeted to mitochondria, it follows that any signal reflects redox conditions in mitochondria.

2) It may be hazardous to rely only on chemicals to study the effects of ROS. Genetic manipulation of ROS by SOD2 (and eventually SOD1 as control), can clarify whether mitochondria are the ROS source and further demonstrate whether ROS is the cause of other phenotypes. As this superoxide dismutase functions in mitochondria, it should quench the generation of ROS induced by spinosad and thus provide an independent, more convincing, demonstration of the role of mitochondria in spinosad-induced toxicity.

3) The authors did not fully address previous issue 3 of reviewer 2, most likely due to a misunderstanding. What I asked for was that the authors check whether spinosad can induce or not behavior defects (bang sensitivity and climbing ability) in adult Dα6 mutants and whether spinosad can cause lipid deposits in the retina of Dα6 mutants. Instead, the authors characterized behavior defects and retina structure in adult Dα6 mutants in the absence of spinosad, which are interesting but incomplete observations in my opinion. In other experiments (e.g. Figure 4D and E, and Figure 5B and G), the authors have indeed compared the effects of spinosad in control and Dα6 mutants. Similarly, I think that it would be worth doing the same test (i.e. check the effects or lack of effects of spinosad on adult Dα6 mutants) for the experiments described in Figure 7A-C and Figure 8A-D.

---

## [Author Response]

[Editors’ note: the authors resubmitted a revised version of the paper for consideration. What follows is the authors’ response to the first round of review.]

Reviewer #2:This article reports that an organic insecticide used at low doses creates nevertheless considerable havoc, including extensive neurodegeneration in adults. The authors show that exposure to spinosad blocks acetylcholine–triggered calcium influx in larval neurons that express the Acetylcholine receptor (AchR) subunit Dalpha6, the target of spinosad. They document that exposure to spinosad leads to a Dalpha6–dependent enlargement of lysosomes in the nervous system where the Dalpha6 subunits accumulates, an increase in reactive oxygen species (ROS) generation in the brain, and a redistribution of lipids in the fat body from large lipid droplets (LDs) to numerous smaller ones, an effect that can be counteracted by pre–exposure to an antioxidant compound. The chronic exposure of adults to low spinosad doses leads to progressively altered behaviors as well as extensive neurodegeneration in the visual system as well as in parts of the central brain.The strengths of the study lie in the description of several damages induced by exposure to spinosad. The part on the neurodegeneration in adults is especially impressive. The central message of the authors that even organic pesticides have unintended severe effects to a nontarget insect upon exposure to low doses (chronic low doses in the case of adults) is thus supported by experimental evidence, which however might be improved in specific cases.The major weaknesses of the study are that the authors hardly investigate causal relationships between the different phenotypes they observe and remain too descriptive. For instance, their preferred model is that lysosomal disorders are at the origin of ROS production in mitochondria, with no experimental evidence to back up this claim. This reviewer is also not thoroughly convinced that ROS are produced by mitochondria, as the decreased activity of aconitase and impact on ATP production might be mediated by another mechanism, e.g., increased mitophagy. They do not consider the alternative possibility that lysosomal dysfunction might be caused by ROS production. Thus, one would like to know whether the expansion of lysosomes still occurs when ROS production is inhibited. This would be best achieved using two independent approaches, one based on exposure to reducing agents (do they have any effect on their own (control missing)?), and the other based on genetic strategies that however require identifying the enzyme(s) responsible for ROS production.They also do not document whether the ROS phenotype depends on the presence of the Dalpha6 subunit of the AchR. It would also be positive to check whether there is any altered production of ROS occurring directly in the fat body or Malpighian tubules. Can the authors exclude that the Dalpha6 nAchR subunit might be expressed at low levels in these tissues, for instance in the enteroendocrine cells of the gut?One major question raised by this study is whether insects other than those targeted by this pesticide are exposed to low doses, for instance in fields neighboring the treated area and undergo damages similar to those reported here. Some of the techniques used in this study may be employed on other insects collected near fields treated with spinosad, such as examining the distribution of LDs in their fat bodies. In this respect, it might be useful determining whether this phenotype as well as the expanded lysosomes observed in larvae are also found in adults.

We thank the reviewer for these comments and have addressed most issues raised (see below).

As stated in above, a major weakness of the study is a lack of functional connections made between the different phenotypes documented by the authors. While it may not be easy to determine whether ROS production depends on lysosome dysfunction, the reverse can be investigated relatively easily, for instance by using larvae pre–treated with NACA and other reagents as detailed below.

1) The levels of superoxide in brains of Dα6 KO mutant flies were measured prior and after spinosad exposure and results added to the manuscript (Figure 4D, E). Text was amended in the following way (lines 222-229): “To test whether the increase of superoxide levels in brains are a consequence of the spinosad induced Dα6 removal from membranes, the levels of superoxide were measured in exposed and non-exposed Dα6 KO mutants. Non-exposed Dα6 KO mutants showed a mild (17%) increase in the levels of superoxide in brains when compared to non-exposed wild type larvae (Figure 4D, E), and exposure to spinosad caused no alteration of superoxide levels in Dα6 KO mutants (Figure 4D, E). Hence, the absence of Dα6 subunit by itself is able to modestly increase the oxidative stress (Weber et al., 2012), but higher levels of ROS are observed in the presence of Dα6 and spinosad.”

2) Experiments exposing NACA pre-treated larvae to spinosad were added. They showed that NACA prevents ROS accumulation (Figure 4F, G) but does not prevent lysosome expansion (Figure 4H, I). Text was amended in the following way (lines 232-238): “Whereas NACA treatment was able to completely prevent ROS accumulation in exposed animals (Figure 4F, G), it does not prevent lysosome expansion (Figure 4H, I). The presence of enlarged lysosomes in the absence of ROS suggests that the onset of lysosomal phenotype is not caused by the rise in oxidative stress levels. NACA, however, reduced the severity of the lysosomal phenotype (mean 1.63% of lysotracker area – Figure 4I, versus mean 2.39% of lysotracker area – Figure 2E). This suggests that, once initiated, the increase in ROS levels may worsen the lysosomal phenotype.”

As regards ROS, one definitely would like to know whether they are generated solely in neurons expressing the Dalpha6 nAchR subunit or whether they are found in other neuronal cell types or glial cells. A subcellular resolution would be ideal. To this end, the authors might want to use ratiometric ROS stress reporters such as Mito–roGFP (Albrechts et al., Cell Metabolism, 2011).

Upon investigation we concluded that Mito-roGFP could not generate the level of resolution expected to indicate the subcellular origin of ROS. Recent data however

(https://www.biorxiv.org/content/10.1101/2021.07.04.451050v1.article-metrics), points that Dα6 is largely expressed in neurons while not expressed in glia, guts, or fat body. Text was amended in the following way (lines 273-275): “No expression of Dα6 has been reported in gut and fat body but it is abundantly and widely expressed in most CNS neurons with little to no expression in glia (Li et al., 2021).”

It would also be desirable to rely on more than one method to block ROS production. In terms of biochemical reducing agents, the authors might want to also check MitoTempo, which is more specifically acting on mitochondrial ROS. On the genetic side, the authors should overexpress Superoxide Dismutase 2 (SOD2), and possibly SOD1 as a control. Another useful tool is a mitochondrially–addressed catalase (UAS–mtcatalase) developed originally by William Orr, e.g., Mockett et al., Free Radic. Biol. Med. , 2003.Once the mitochondrial origin of ROS is demonstrated, it will then be highly interesting to determine whether lysosome expansion still occurs when the source is inhibited.

In response to spinosad exposure we show: (a) accumulation of superoxide – main ROS produced by mitochondria – in brain and guts (Figure 4A, B); (b) reduction of mitochondrial aconitase activity (Figure 3D) and (c) ATP (Figure 3E); (d) increased mitochondrial turnover (MitoTimer) (Figure 3A, B); and (e) reduced cardiolipin levels (Figure 6C). These data strongly argue that the observed ROS in wild-type exposed flies is likely not due to a xenobiotic response. However, we cannot rule out that this response does not play a minor role. We further demonstrate that NACA rescues improves motility (Figure 3F) and survival (Figure 3G) of spinosad exposed larvae, while reducing impacts on the lipid environment of metabolic tissues (Figure 5C, D). One of the novelties present in this work is connecting the generation of ROS to spinosad mode of action.

The authors rely solely on Nile Red for assessing the distribution of LDs. A control with lipidTox staining, which is more specific for neutral lipids, would be appropriate. Also, while small LDs may increase the surface available for storing oxidized lipids, it would be useful to determine whether this is indeed the case by staining oxidized lipids.

Microscopy with Nile red securely shows the increase in lipid droplets after spinosad exposure (Figure 5A, B) and the amelioration of this phenotype with NACA pre-treatment (Figure 5C, D). Despite LipiTox higher affinity for neutral lipids, experiments with this staining would not change the lipid droplet count obtained with Nile red.

It is also not clear whether the overall levels of triglycerides are changed upon spinosad exposure and measuring the levels of TAGs biochemically would provide an interesting counterpoint to lipid stains.

We have observed the increase of some TAG species in our lipidomic profiles, as well as the decrease of others (Figure 6A; Figure 6 – table supplement 1). While the use of whole larvae for lipidomic analysis reduced the capacity to detect significant shifts in lipid levels that predominantly occur in individual tissues, it allowed the identification of broader impacts on larval biology. The levels of circulating lipids in hemolymph as well as the lipid droplet counts were assessed in two wild-type strains and their respective Dα6 KO mutants. In all cases, increased lipid levels were found in response to spinosad exposure in wild-type flies (Figure 5A, B and G). And whereas Dα6 KO mutants present higher lipid levels than their wild-type strains, such levels did not change after spinosad exposure (Figure 5A, B and G).

What is the biological function of this redistribution of lipids in the fat body? Would the genetic inhibition of Dalpha6–expressing neurons lead to the LD phenotype (likely unlikely given the loss–of–function phenotype). Conversely, would genetically activating neuronal signaling in these neurons be able to reverse the effects of spinosad exposure, at least as regards the LD phenotype in the fat body?

The knockout of Dα6 leads to mild alterations in the lipid environment. It increases lipid stores in fat body and lipid levels in hemolymph (Figure 5A, B and G). Dα6 loss of function mutation has been linked to an increased susceptibility to oxidative stress

(DOI: 10.1371/journal.pone.0034745). Many sections in the manuscript were restructured to clearly point the difference between phenotypes causes by spinosad exposure and those caused by Dα6 loss of function. Regarding genetically activation of neuron signalling, since nicotinic receptors are involved in a plethora of functions (from muscle activity to hormone levels control and behaviour), the activation of neural signalling would have a diverse range of significant effects that should first be addressed before testing its effect on spinosad exposure.

Reviewer #3:In this paper, the authors used *Drosophila* as a model to study the effect of the insecticide spinosad on the non–target insect. They investigated the histological, physiological, and behavioral impacts of spinosad and found that low dose administration of this pesticide causes neuronal lysosomal alterations and mitochondrial impairment and ROS elevation and lead to altered lipids profile and neurodegeneration and subsequent consequences. One of the main results of the study is to show that spinosad induce major systemic metabolic changes through its action in the brain. Since spinosad have been shown to greatly affect the non–pest insect population, the findings here are interesting and appealing to many researchers. I only have two major comments as listed below.1. The authors suggest that spinosad actions on neurons expressing Dα6 nAChRs inducing subsequent lipid phenotypes, especially through ROS. They also observed an increased ROS level in tissues as gut. Although Dα6 mutant showed a decreased lipid droplets in fatbody, there is no direct evidence linking neuronal ROS with this phenotype. Thus, it remains unclear whether peripheral tissues ROS also contribute to the altered lipid phenotypes in the fly. Furthermore, whether elevated gut ROS level is induced by neuronal ROS burst or independent from neuronal phenotype is not clear. I would suggest using fly genetic tools (Gal4–uas system) to manipulate ROS level in the neurons and peripheral tissues to further elucidate the conclusion.2. The author's finding pointed to the mechanisms of spinosad on non–target insects. However, since insects are the most diverse group of animals and there is a huge divergence between *Drosophila* and other insects, whether spinosad causes the same effects in other species needs to be clearly discussed.

We thank the reviewer for these comments and have addressed most issues raised (see below).

3. The authors suggest that the spinosad effect on *Drosophila* was a consequence of its action on Dα6 nAChRs. It would be better to include survival data of Dα6 mutant upon Spinosad exposure to reinforce the conclusion.

Data showing the survival of Dα6 mutant upon spinosad exposure was added (Figure 1C). Text was amended in the following way (lines 117-119): “Under this exposure condition, only 4% of wild type larvae survived to adulthood (Figure 1B), whereas 88% nAChRα6 knockout (Dα6 KO) mutants survived (Figure 1C)”.

4. It would also be interesting to see if there is also a ROS burst in the fatbody that leads to the metabolic phenotype directly.

Data showing that no burst of superoxide is observed in fat body upon spinosad exposure was added (Figure 5E, F). Text was amended in the following way (lines 286-287): “However, measurements of superoxide levels in fat bodies showed no differences between spinosad exposed and non-exposed larvae (Figure 5E, F).”.

Overexpression of antioxidant enzymes (sod1, sod2, catalase) using elav–Gal4 (Pan neuron driver), myo–Gal4 (gut enterocytes driver) and c564–Gal4 (fatbody driver) could elucidate the causal relations between the neuronal phenotypes and lipid metabolic phenotypes.

We do not exclude the possibility that oxidative stress is directly generated in the midgut but in a previous work published in PNAS (Martelli et al.,2020) we demonstrate that knocking out mitochondrial genes (ND42 and Marf) in the nervous system (elav-Gal4) increases oxidative stress in the brain and, also lead to an accumulation of lipid droplets in fat body as well as a reduction of lipid droplets in Malpighian tubules. The text was amended to emphasised this (lines 257-265): “Oxidative stress has the ability to affect the lipid environment of metabolic tissues, causing bulk redistribution of lipids into lipid droplets (LD) (Bailey et al., 2015). The RNAi knockdown of mitochondrial genes, Marf and ND42, in *Drosophila* neurons was shown to increase the levels of ROS in the brain and precipitate the accumulation of LD in glial cells (Liu et al., 2015). Martelli et al. (2020) showed that the knockdown of the same mitochondrial genes in *Drosophila* neurons can also precipitate the accumulation of LD in fat bodies and a reduction of LD in Malpighian tubules. Such changes in the lipid environment of metabolic tissues were recapitulated by low imidacloprid exposures, which like spinosad, also causes an increase of ROS levels in the brain that further spreads to the anterior midgut (Martelli et al., 2020).”

Reviewer #4:The manuscript by Felipe Martelli and coll. addresses the cellular and metabolic defects induced by poisoning with spinosad in the fruit fly *Drosophila*. Spinosad is a potent insecticide first identified as being produced by a soil bacterium and so considered as an organic product, which has been widely used in recent years. Published work showed that the main selective target of spinosad in insects is the α6 subunit of the nicotinic acetylcholine receptor (nAchR), which is selectively expressed in the nervous system in flies, and that knockout of nAChRα6 (Dα6) in *Drosophila* confers resistance to spinosad. This paper complements and extends these previous studies by showing that spinosad exposure triggers lysosomal swelling, mitochondrial impairment and oxidative stress in fly larval brain cells, and also induces a variety of metabolic dysregulation in peripheral tissues, including alterations in the larval lipidomic profile and an accumulation of small lipid droplets in the fat body. Pretreatment with the anti–oxidant N–acetylcysteine amide (NACA) alleviated some of these defects and partially, the larval lethality induced by spinosad. It is also shown that low doses of spinosad promoted neurodegeneration and loss of vision in adult female *Drosophila*.Overall, this manuscript represents a substantial amount of experimental work that provides new insight into spinosad–induced toxicity. The authors introduce here several hypotheses to make sense of their various observations. Specifically, they propose that spinosad accumulates in lysosomes after binding to nAChRα6 and because of that would progressively disrupt lysosomal function. This could then give rise to mitochondrial dysfunction, increased ROS production and finally neurodegeneration. The authors also suggest that a toxic signal released from the brain after spinosad intoxication propagates to peripheral tissues, including the fat body, midgut, and Malpighian tubules, to promote disturbance in metabolism and lipid homeostasis. Although these hypotheses are not directly addressed experimentally in this study, they would certainly be interesting to explore in future work.There are three sets of results for which I have concerns about some of the authors' conclusions:– In Figure 1, it is not clear whether the lower carbachol–induced response in spinosad–treated cells reflects a decrease in calcium influx through nAChRs, or, alternatively, a reduced calcium release from intracellular stores (see e.g. Campusano et al. Dev Neurobiol 2007). This point has to be carefully considered as the lower response to carbachol could be either a direct or indirect effect of spinosad–induced Dα6 subunit internalization. I think that the reported experiments are not sufficient to decide between these possibilities.– The second issue is the experiment of Figure 5, in which the authors showed that spinosad treatment of Dα6 KO flies had no effect on lipid level in the hemolymph, in contrast to the increase it induced in wild–type controls. This conclusion is apparently correct, but it does not take in account the fact that the lipid level actually appears at least as high in the Dα6 mutants as in their respective controls treated with spinosad. It seems therefore that the increase in lipid in the hemolymph is more likely caused by Dα6 deficiency than by spinosad treatment. Similarly, the percentage of the area occupied by LD in fat body of both Dα6 mutants is between that of untreated control flies and control flies treated with spinosad. This seems to be consistent with the idea that spinosad only exacerbates an increase in lipids which is for a large part induced by Dα6 deficiency. Incidentally, because Dα6 KO mutants are viable, this observation indicates that altered lipid homeostasis may not be a prominent cause of larval lethality in spinosad–exposed flies.– My third main comment is that it would be very informative to compare the effects of spinosad in adult wild–type and Dα6 mutant flies, with respect to the survival, bang sensitivity and climbing ability tests (Figure 7), as well as for lipid deposits in the retina (Figure 8). These data would be rather easy to collect and would usefully complete these figures by indicating whether or not spinosad can induce defects in adult flies in the absence of Dα6. A positive answer would argue for the existence of other potential molecular targets of this insecticide.I have provided more detail on these and other points below.

We thank the reviewer for these comments and have addressed most issues raised (see below).

Main issues1) Line 117 and 132: from experiments shown in Figure 1D, E, the authors conclude that spinosad exposure "prevent ca^2+^ flux into neurons expressing nAChRs" and so "blocks the function of Dα6–containing nAChRs". However, Campusano et al. Dev Neurobiol 2007, have shown that thapsigargin treatment reduces nicotine‐evoked calcium increase in Kenyon cells, consistent with a signal amplification by calcium release from intracellular stores. In addition, part of this calcium increase had a voltage‐gated calcium channel–dependent (VGCC) component, indicating that only a fraction of the calcium increase involves calcium influx directly through nAChRs. Therefore, the experiments in Figure 1D, E are not sufficient to decide, in my opinion, whether the reduction in carbachol–induced response in spinosad–treated cells reflects a decrease in calcium influx through nAChRs, or, alternatively, a reduced calcium release from intracellular stores. As mentioned in the public review, this point has to be carefully considered as the decreased response to carbachol could be either a direct or indirect effect of spinosad–induced Dα6 subunit internalization.

The reviewer is correct to state that with the current results it is not possible to determine whether the ca^2+^ transients reflect reduced influx from internal or external sources. What is clear, however, is that spinosad exposure led to a diminished ca^2+^ transient. No evidence for stimulus is observed. That is an important observation since past publications would assume that spinosad could provoke a stimulating action on Dα6 receptors. The text was amended in the following way (lines 126-130): “While it was not determined whether the ca^2+^ transients reflect reduced influx from internal or external sources (Campusano et al., 2007), spinosad exposure led to a diminished ca^2+^ transient and reduced cholinergic response. Hence, in contrast to imidacloprid, which leads to an enduring ca^2+^ influx in neurons, spinosad reduces the ca^2+^ response mediated by Dα6 (Martelli et al., 2020).”

2) Line 263: "We also quantified the level of lipids in hemolymph. Whereas Line 14 and Canton S showed an average 10% and 13% increase in response to spinosad, respectively, neither of the Dα6 KO mutants showed significant changes (Figure 5E)." However, the lipid level appears to be at least as high in the Dα6 mutants as in their respective controls treated by spinosad (Figure 5E). It seems therefore that the increase in lipid in the hemolymph is more likely caused by Dα6 deficiency than by spinosad treatment. Similarly, the percentage of area occupied by LD in fat body of both Dα6 mutants is between that of untreated control flies and control flies treated with spinosad (compare Figure 4B and Figure 5B for the Line 14 background, and compare Figure 5C and Figure 5D for the Canton S background). This seems to be consistent with the idea that spinosad only exacerbates an increase in lipids which is largely induced by Dα6 deficiency. If this is correct, please amend the conclusions accordingly. For example, the sentence on page 25, line 479: "Dα6 knockout mutants exposed to spinosad show no accumulation of LD in the fat body or change of lipid levels in hemolymph indicating that these phenotypes are due to the spinosad:Dα6 interaction (Figure 5)" is not fully correct as it does not mention the fact that Dα6 deficiency by itself appears to increase LD accumulation.

Many sections in the manuscript were restructured, and new datasets added to better characterize the phenotypes caused by Dα6 loss of function mutation and distinguish those from the ones caused by spinosad:Dα6 interaction. While phenotypes in mutants are mild compared to the ones induced by exposure, they are not neglectable and are here for the first time characterized.

3) It would be very informative to repeat the survival and behavior test experiments of Figure 7 in Dα6 mutant background. I was actually surprised that this was not included in the manuscript, as it would not involve a lot of work, and, importantly, it would indicate whether spinosad can induce behavior defects in the absence of Dα6, which would argue for the existence of other molecular targets for this toxin. I would recommend that lipid deposits in the retina (Figure 8A, B) be also quantified in Dα6 background for purpose of comparison.

Data characterizing the survivorship, climbing ability, bang sensitivity (Figure 7D-F) and morphology of retinas (Figure 8E, F) of Dα6 mutants were all added to the manuscript.

– Line 579: "derived from Armenia60 (*Drosophila* Genomics Resource Center #103394)" – I could not find this Armenia60 line referenced neither in the DGRC or Flybase sites. Please provide the correct source or link for this strain, and explain why it was used as a wild–type control.

Armenia60 is a strain received from the Umea Stock Center in 2001, this centre is now closed however the strain was transferred to DGRC and is called Aashtrak (DGRC #103394). https://kyotofly.kit.jp/cgi-bin/stocks/search_res_det.cgi?DB_NUM=1&DG_NUM=103394. It has been used in several studies as both a neonicotinoid susceptible strain (Perry et al. 2008) and a spinosyn susceptible strain (Somers et al. 2015). Line 14 refers to a specific isofemale derived strain of Armenia60 which do not overexpress the P450 gene Cyp6g1 (Perry et al. 2008), a known mechanism of resistance against neonicotinoids.

– Line 125: "primary culture of neurons " could be "primary culture of third–instar larva brain neurons" to avoid misunderstanding.– In Figure 1A, the statistical test used should be two–way ANOVA and not t–test, I think, as both the genotypes and time differ in each point, and in Figure 1E one–way ANOVA should be used, as the means of more than two groups were compared. Same issue in Figure 2B and in several other figures of the paper where the Student's t–test was systematically used.– Figure 1B: I understand that spinosad precludes pupation of intoxicated larvae. However, after eight days, a wild–type third–instar larva should have become an adult fly. So, I guess that authors looked at fly survival, and not larval survival, for the wild–type. Please mention it, if it is right. In Figure 1C, did you treat the pupae or the larvae before pupal case formation with the toxin? This could be stated in the legend.– Line 581: "Expression of nAChR–Dα6 gene" – According to Flybase, this gene should be named: nAChRα6. The correct nomenclature should be mentioned at least once in the manuscript.– Line 582: "the MitoTimer line (Gottlieb and Stotland, 2015) was used". The exact reference for this line is: Laker et al. J Biol Chem. (2014) 289(17):12005–15. Same correction required on line 187.– Line 594: "In all experiments involving adult flies only females were used to maintain consistency." Adult *Drosophila* females have very different physiology compared to males. This is particularly relevant for reproductive organs, of course, but also for the gut and nervous system. Therefore, this particularity of this study is not negligible and should be mentioned in the abstract.

[Editors’ note: what follows is the authors’ response to the second round of review.]

Essential revisions:1) While this revised version is definitely improved, a sticky point remains, namely that of the relationship between the lysosomal disorder and the production of ROS by mitochondria, which is not rigorously established. The authors use NACA to block the production of ROS. Since presumably NACA has a ubiquitous effect, the authors cannot formally exclude that the ROS originate from a source distinct from mitochondria. In this respect, it is noteworthy that the authors did not investigate the impact of NACA treatment on mitochondria. This reviewer does not understand the argument put forth by the authors as regards the lack of subcellular resolution of mito-roGFP reporters. Because the reporters are targeted to mitochondria, it follows that any signal reflects redox conditions in mitochondria.

Following the reviewer’s suggestion we used the mito-roGFP2-Orp1 strain (BDSC #67672) to investigate the mitochondrial origin of ROS. Figure 3 —figure supplement 1 was included, and the text was amended in the following way (lines 185-192): “The mito-roGFP2-Orp1 construct, an in vivo mitochondrial H2O2 reporter (Albrecht et al., 2011), was used to identify the origin of ROS induced by spinosad exposure at 2.5 ppm for 2 hrs. A subtle, but significant increase in the 405 (oxidized mitochondria signal)/488 (reduced mitochondria signal) ratio was observed in the brain (20% on average) and anterior midgut (10% on average) (Figure 3 —figure supplement 1), pointing to a rise in H2O2 generation upon a few hours of exposure. Similarly, to the MitoTimer reporter, an increase in the oxidized mitochondrial signal was accompanied by the increase in the reduced mitochondrial signal, accounting for the subtle increase in 405/488 ratio.”

2) It may be hazardous to rely only on chemicals to study the effects of ROS. Genetic manipulation of ROS by SOD2 (and eventually SOD1 as control), can clarify whether mitochondria are the ROS source and further demonstrate whether ROS is the cause of other phenotypes. As this superoxide dismutase functions in mitochondria, it should quench the generation of ROS induced by spinosad and thus provide an independent, more convincing, demonstration of the role of mitochondria in spinosad-induced toxicity.

Following the reviewer’s suggestion we used an *elav-GAL4* driver (BDSC #458) to express *Sod2* (BDSC #24494) and *Sod1* (BDSC #24750) in the nervous system and investigate the mitochondrial origin of ROS. Figure 4D, E was included, and the text was amended in the following way (lines 248-254): “To assess the mitochondrial origin of the ROS measured with DHE, flies expressing the superoxide dismutase gene *Sod2* in the nervous system with the *elav-GAL4* driver were exposed to 2.5 ppm spinosad for 2 hr. Sod2 is the main ROS scavenger in *Drosophila* and is localized to mitochondria (Missirlis et al., 2003). Sod1 is present in the cytosol (Missirlis et al., 2003), and expression of this gene was used as a control. While exposure to spinosad caused an average 63% increase in ROS levels in control larvae overexpressing *Sod1*, an average increase of only 28% was found in larvae overexpressing *Sod2* (Figure 4D, E).”. We also added (lines 569-576): “While we cannot rule out the generation of ROS by other mechanisms, we provide compelling evidence that a significant part of ROS that is generated by spinosad exposure is of mitochondrial origin, arguing that mitochondrial impairment is a key element of spinosad mode of action at low dose exposure. The evidence is based on increased mitochondrial turnover and mito-roGFP 405/488 ratio, reduced activity of the ROS sensitive enzyme m-aconitase and reduced ATP levels (Figure 3). In addition, we observed a highly significant reduction of cardiolipin levels (Figure 6C) typically associated with defects in the electron transport chain and increased ROS production as they are required for the anchoring of Complex1 in mitochondria (Dudek, 2017; Quintana et al., 2010).”

3) The authors did not fully address previous issue 3 of reviewer 2, most likely due to a misunderstanding. What I asked for was that the authors check whether spinosad can induce or not behavior defects (bang sensitivity and climbing ability) in adult Dα6 mutants and whether spinosad can cause lipid deposits in the retina of Dα6 mutants. Instead, the authors characterized behavior defects and retina structure in adult Dα6 mutants in the absence of spinosad, which are interesting but incomplete observations in my opinion. In other experiments (e.g. Figure 4D and E, and Figure 5B and G), the authors have indeed compared the effects of spinosad in control and Dα6 mutants. Similarly, I think that it would be worth doing the same test (i.e. check the effects or lack of effects of spinosad on adult Dα6 mutants) for the experiments described in Figure 7A-C and Figure 8A-D.

Following the reviewer’s suggestion we have now included experiments to assess the effects of spinosad exposure on viability and behavioral phenotypes of *Dα6* mutant flies. Figure 7E-G was included, and the text was amended in the following way (lines 395-408): “The same phenotypes were also assessed in adult female virgin *Dα6 KO* mutants. Unexposed mutant flies show a significant reduction in longevity compared to unexposed wild-type individuals, but that difference is only noticeable later in life, Canton-S *Dα6 KO* mutants have a median survival of 68 days compared to 82 for Canton-S (Figure 7D). A 25 day exposure to 0.2 ppm spinosad leads to a 91% survival of Canton-S *Dα6 KO* mutants whereas only 40% of Canton-S wild-type flies survive this exposure (Figure 7E). No changes in bang sensitivity or climbing ability were observed between exposed and unexposed *Dα6 KO* mutants over the course of a 20 day exposure (Figure 7F, G). However, at the 20 day time-point, twice as many of unexposed *Dα6 KO* mutants failed to climb (36%) compared to unexposed Canton-S wild-type flies (18%). (Figure 7G). These data support that the deleterious effects of spinosad are mediated by its binding to *Dα6* receptors. They also indicate that loss of *Dα6* by itself causes mild but significant behavioral and lifespan phenotypes not previously reported.”